# Light-triggered switching of liposome surface charge directs delivery of membrane impermeable payloads in vivo

Gabriela Arias-Alpizar [1,7], Li Kong[1,2,7], Redmar C. Vlieg[3], Alexander Rabe [4,6], Panagiota Papadopoulou [1], Michael S. Meijer [5], Sylvestre Bonnet [5], Stefan Vogel [4], John van Noort [3], Alexander Kros [1✉] & Frederick Campbell [1✉]

Surface charge plays a fundamental role in determining the fate of a nanoparticle, and any encapsulated contents, in vivo. Herein, we describe, and visualise in real time, light-triggered switching of liposome surface charge, from neutral to cationic, in situ and in vivo (embryonic zebrafish). Prior to light activation, intravenously administered liposomes, composed of just two lipid reagents, freely circulate and successfully evade innate immune cells present in the fish. Upon in situ irradiation and surface charge switching, however, liposomes rapidly adsorb to, and are taken up by, endothelial cells and/or are phagocytosed by blood resident macrophages. Coupling complete external control of nanoparticle targeting together with the intracellular delivery of encapsulated (and membrane impermeable) cargos, these compositionally simple liposomes are proof that advanced nanoparticle function in vivo does not require increased design complexity but rather a thorough understanding of the fundamental nano-bio interactions involved.

[1] Department o Supramolecular & Biomaterials Chemistry, Leiden Institute of Chemistry (LIC), Leiden University, P.O. Box 9502, 2300 RA, Leiden, The Netherlands. [2] Tongji School of Pharmacy, Huazhong University of Science and Technology, 430030 Wuhan, P.R. China. [3] Leiden Institute of Physics (LION), Leiden University, P.O. Box 9504, 2300 RA, Leiden, The Netherlands. [4] Department of Physics, Chemistry and Pharmacy, University of Southern Denmark, 5230 Odense, Denmark. [5] Department of Metals in Catalysis, Biomimetics & Inorganic Materials (MCBIM), Leiden Institute of Chemistry (LIC), Leiden University, P.O. Box 9502, 2300 RA, Leiden, The Netherlands. [6] Present address: BioNTech RNA Pharmaceuticals GmbH, An der Goldgrube 12, 55131 Mainz, Germany. [7] These authors contributed equally: Gabriela Arias-Alpizar, Li Kong. ✉email: a.kros@chem.leidenuniv.nl; f.campbell@lic.leidenuniv.nl

Surface charge is a key determinant of nanoparticle fate in vivo[1,2]. Following intravenous (i.v.) injection, nanoparticles with high surface charge density, either anionic or cationic, are rapidly cleared from circulation by specialised cells of the reticulo-endothelial system (RES)[3–5]. In mammals, RES cell types are primarily located in the liver (key hepatic RES cell types: Kupffer cells, KCs, and liver sinusoidal endothelial cells, LSECs) and spleen. These cells are responsible for clearing up to 99% of i.v. administered nanoparticles from circulation[6]. High nanoparticle surface charge density has a qualitative and quantitative impact on serum protein binding[7–12], driving the opsonisation of circulating nanoparticles and subsequent recognition and clearance by the RES[13–15]. In addition, cationic nanoparticles tend to adsorb to the anionic surface of cells and are subsequently internalised[16–20], often leading to acute cytotoxicity[21–23]. Given the adverse pharmacokinetics of charged nanoparticles in the body, most clinically approved, nanoparticle-drug formulations (nanomedicines) possess a (near) neutral surface charge to prolong circulation lifetimes and maximise drug exposure within target (vascularized) tissues in the body[24].

We have previously shown that *i.v.* administered liposomes with (near) neutral surface charge, and optimally 100 nm in size, tend to freely circulate in embryonic zebrafish (*Danio rerio*)[25]. Anionic nanoparticles (in our experience, <−20 mV measured zeta (ζ) potential) interact strongly with RES cell types, namely scavenging endothelial cells (SECs, via a *stabilin*-mediated clearance pathway) and blood resident macrophages[25,26]. Whereas, cationic liposomes (>20 mV measured ζ-potential) are rapidly removed from circulation through a combination of non-specific cellular interactions (i.e. adsorption to the anionic surface of the blood vessel walls), and/or clearance via the RES[25]. While usually considered detrimental to in vivo performance, the non-specific, cellular interactions of cationic nanoparticles/complexes (e.g. Lipofectamine™) have been widely exploited to deliver membrane impermeable, (genetic) material across cell membranes in vitro[27–30]. In these cases, a net cationic surface charge not only promotes non-specific adsorption and uptake within cultured cells but also facilitates endosomal escape and cytosolic payload release. In contrast, anionic and neutral nanoparticles are generally taken up sparingly by non-RES cell types, while those that are internalised tend to localise within lysosomes[21,31]—a chemically hostile environment in which encapsulated payloads are rapidly degraded.

The contrasting fates of differently charged nanoparticles have all the ingredients of an ideal targeted drug delivery system. On the one hand, i.v. administered, (near) neutral nanoparticles freely circulate, maximising exposure within any (vascularized) tissue of the body. On the other, cationic nanoparticles are non-specifically taken up by virtually all cells, delivering high intracellular concentrations of encapsulated (and membrane impermeable) payloads. Herein, we describe the rapid switching of liposome surface charge, from neutral to cationic, in situ and in vivo using light as exclusive trigger (Fig. 1). Light is chosen as trigger given the rapid and quantitative photolysis of common chemical photocages[32], its proven clinical relevance[33] and the prospect of emerging technologies to apply light deep within patients. These include fiber-optic[34,35] and injectable microLED hardware[36], as well as photocleavable chemical functionality sensitive to visible or near infrared (NIR) light[37–39]. Light wavelengths between 600 and 950 nm can penetrate various human tissues (skin, fat and blood) up to a depth of 2 cm[40]. As model organism, we select the small and transparent zebrafish embryo. This organism is increasingly being used as a versatile preclinical screening platform for nanoparticles[41] and offers unprecedented opportunities to image nanoparticles across whole live organisms (i.e. visualising total injected nanoparticle doses),

at cellular resolution and in real time[42]. Moreover, the zebrafish embryo can qualitatively predict nanoparticle interactions with scavenging cell types of the RES in mammalian models[25,43].

In this study, following i.v. administration within a zebrafish embryo, photoactive liposomes, composed of just two lipids and prior to light activation, freely circulate and do not significantly interact with RES and/or other cell types of the embryo. Following in situ light activation, however, rapid surface charge switching results in non-specific adsorption and uptake of liposomes across the entire endothelium of the fish, as well as phagocytic uptake in blood resident macrophages. Importantly, light triggered surface charge switching does not disrupt liposome integrity and encapsulated, membrane impermeable payloads are successfully transported across cell membranes following surface charge switching.

## Results

**Design of photoactive liposomes.** Photoswitching the surface charge of a liposome—from neutral to cationic—requires photoactive lipids embedded within a liposome membrane (Fig. 1). In the absence of light and at physiological pH, photocaged lipids should carry no net charge to maintain a (near) neutral liposome surface charge (i.e. freely circulating). To ensure sufficient cationic surface charge density following photoactivation (in our experience, liposomes with a measured ζ-potential >20 mV), photocaged lipids should make up a significant proportion, if not all, of the overall lipid membrane composition. And, for optimal performance, photolysis and subsequent charge switching should be rapid. Finally, to achieve intracellular delivery of (membrane impermeable) drugs, encapsulated payloads should remain entrapped within liposomes, before, during and after light activation. Surface charge switching should not, therefore, involve any large-scale reorganisation of the liposome membrane and with it the potential for leakage of encapsulated drugs.

To ensure the non-specific adsorption of cationic liposomes to blood vessel walls following light triggered surface charge switching, we first assessed the physicochemical properties and in vivo behaviour of liposomes containing cationic, cholesterylamine compounds, **1-3** (Fig. 2a, see Supplementary Information for synthesis and characterisation). Cholesterol can be incorporated into a reconstituted (phospho)lipid bilayer up to ~50 mol%[44], and is often included in liposomal formulations to modulate drug retention and release profiles[45]. Knowing the hydroxyl head group of cholesterol sits deeper within a lipid bilayer than neighbouring phospholipid head groups[46], a series of cholesterylamine compounds, **1-3**, were assessed, in which the spacer length between the hydrophobic cholesteryl anchor and primary amine head group was varied. In all cases, linkers were connected to cholesterol via an ester bond. While spacers were primarily included to ensure effective charge presentation at the lipid-water interface, our choice of spacer chemistry was also influenced knowing the final photocaged cholesteryl compounds would be charge neutral, hydrophobic and potentially form lipid droplets within a phospholipid membrane[47]. In this scenario and upon light activation, we envisaged extensive membrane remodelling to reposition the newly revealed primary amine at the water-lipid interface and with it the potential for contents leakage. To minimise this risk and to increase the amphipathicity of the final photocaged cholesteryl compound, we focused on short glycine and polyethylene glycol (PEG) linkers as hydrophilic and/or uncharged spacers.

Liposomes, containing varying amounts of 1-3, up to 50 mol%, were co-formulated with zwitterionic, 1,2-dioleoyl-*sn*-glycero-3-phosphocholine (DOPC) phospholipids. All formulated liposomes were prepared by standard extrusion techniques and were

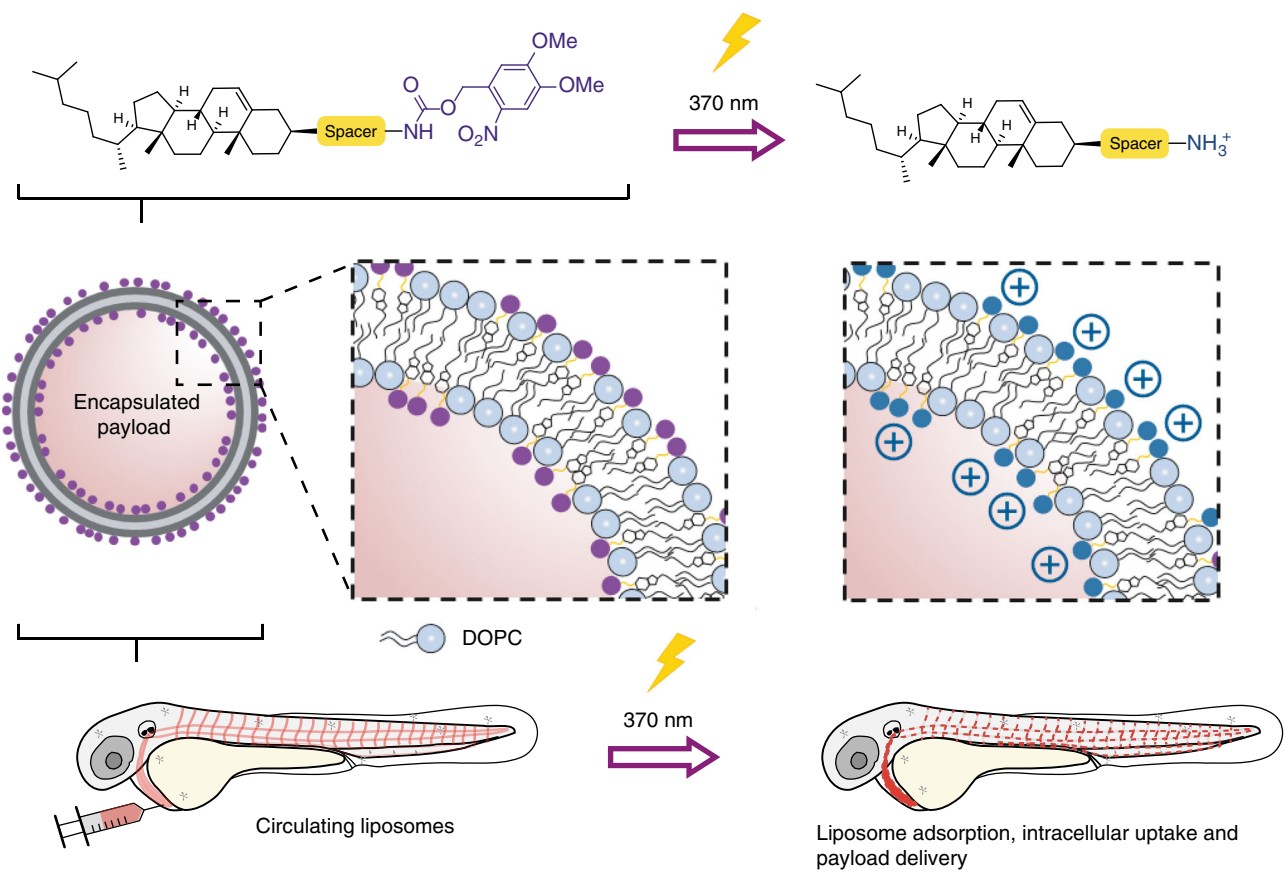

**Fig. 1 Photoswitching the surface charge of a liposome – from neutral to cationic – requires photoactive lipids embedded within a liposome membrane.** Prior to light activation, charge neutral, photoactive liposomes freely circulate throughout the vasculature of a zebrafish embryo and do not interact with RES cell types, or any other cell type, of the embryonic fish. Upon light irradiation and photolysis of photocaged, cholesterylamine lipids, rapid surface charge switching, from neutral to cationic, leads to non-specific adsorption of liposomes across the endothelium of the embryo, liposome uptake and intracellular delivery of liposome-encapsulated, membrane impermeable payloads. DOPC 1,2-dioleoyl-sn-glycero-3-phosphocholine.

~100 nm in size, with polydispersity indices (PDI) < 0.2 as measured by dynamic light scattering (DLS, Supplementary Table 1). DOPC was chosen as co-formulant phospholipid as we have previously shown liposomes composed of 100% DOPC[25], as well as 1:1 mixtures of DOPC and cholesterol (Supplementary Figure 1), freely circulate throughout the vasculature of an embryonic zebrafish beyond 1 h post-injection (hpi). As expected, increasing the amount of cholesterylamine, 1-3, within the DOPC liposome membrane resulted in greater cationic surface charge (Fig. 2b and Supplementary Table 1). However, at high mol% of cationic lipids, a trend emerged whereby longer spacers resulted in an increasingly cationic surface charge. Satisfyingly, the measured surface charge of DOPC:3 (1:1) liposomes was comparable to liposomes formulated with commercially available, cationic 1,2-dioleoyl-3-trimethylammonium-propane (DOTAP) lipids (Fig. 2b). We have previously shown that i.v. administered DOTAP liposomes (e.g. EndoTAG-1® - DOPC:DOTAP (45:55) and 100% DOTAP liposomes) non-specifically adsorb to blood vessel walls throughout the vasculature of an embryonic zebrafish[25]. Following i.v. microinjection in a two day old zebrafish embryo (2 days post-fertilisation, dpf), all three cationic liposome formulations—i.e. DOPC co-formulated with 50 mol% 1, 2 or 3—showed comparable biodistributions to cationic DOPC:DOTAP (1:1) liposomes (Supplementary Fig. 2a). In all cases, liposomes were mainly visible as immobile punctae bound to all blood vessel walls (both arterial and venous) and largely removed from circulation at 1 hpi (Fig. 3a–c and Supplementary Fig. 2b, c). In contrast, DOPC liposomes co-formulated with lower mol% of

cholesterylamine 3 showed variable biodistributions dependent on the surface charge density of the liposome (Fig. 3d,i). In particular, (near) neutral DOPC liposomes, containing 10 mol% 3, were extensively taken up by blood-resident macrophages within the caudal haematopoietic tissue (CHT) of the embryonic fish (Fig. 3j,k and Supplementary Fig. 3 for whole-embryo images).

**Light-triggered switching of liposome surface charge in vitro and in vivo.** As DOPC:3 (1:1) liposomes possessed the highest measured cationic surface charge, we proceeded to photocage 3, forming the uncharged, photoactive cholesteryl compound, 4 (Fig. 4a, see Supplementary Information for synthesis and characterisation). Upon UV irradiation (370 ± 7 nm, 202 mW cm$^{-2}$), in H$_2$O/MeCN/$^t$BuOH (1:1:1), complete photolysis of 4 was achieved in less than two minutes, with clean photolysis confirmed by the appearance of two isosbestic points in the UV spectra (Supplementary Fig. 4a). Irradiation of DOPC:4 (1:1) liposomes, formulated in 10 mM HEPES buffer (pH 7.4), resulted in comprehensive surface charge reversal—from ζ-potential −8 to +26 mV—within this same short timeframe (Fig. 4b). Despite batch-to-batch variation (resulting in measured zeta potentials ranging from +20 to +35 mV ζ-potential), the cationic surface charge of irradiated DOPC:4 → 3 (1:1) liposomes was consistently lower than that of parent DOPC:3 (1:1) liposomes (ζ-potential +48 mV). Both formulations should, in theory, be compositionally identical and, at this point, we do not have a

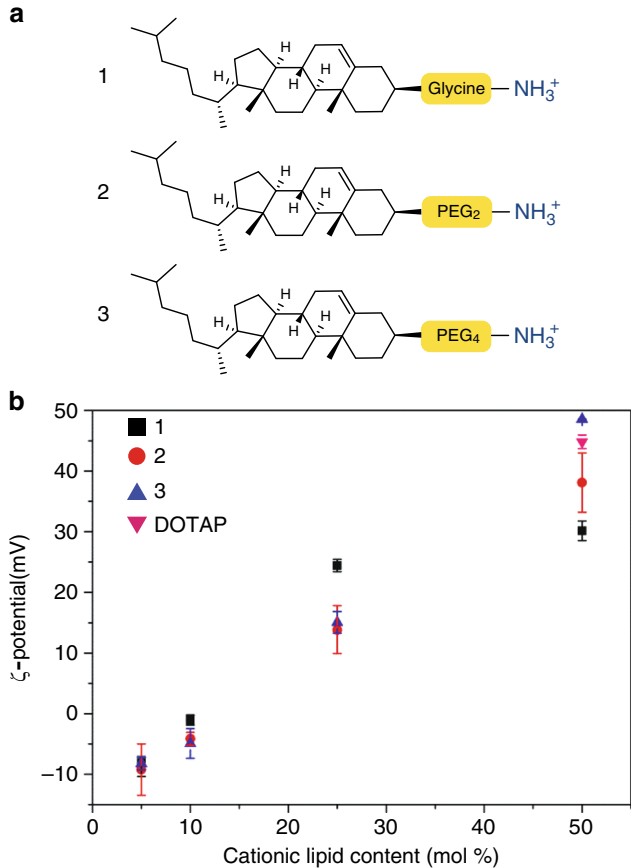

**Fig. 2 Varying cholesterylamine content within DOPC liposomes.**
**a** Cationic cholesterylamine analogues, **1-3**. **b** Measured surface charges (ζ-potential) of DOPC liposomes containing varying mol% **1-3**. DOPC: DOTAP (1:1) liposomes are included as a representative (commercially available) cationic liposome formulation. Measure of centre: mean; Error bars: standard deviation.

reasonable explanation for this discrepancy. The measured size (approx. 100 nm) and PDI (< 0.1) of DOPC:**4** (1:1) liposomes was unchanged before and after irradiation (Supplementary Figure 4b and Supplementary Table 1). Likewise, their spherical, unilamellar morphology, as imaged by cryoTEM, generally remained unchanged before and after UV irradiation (Fig. 4c, see Supplementary Fig. 5 for larger populations), although a small fraction (<10%) of irradiated DOPC:**4** → **3** liposomes did appear non-spherical (oblong) (Supplementary Fig. 5b, e). Whether this morphological change is the result of membrane reorganization upon light triggered photolysis of membrane embedded **4** is hard to conclude given we persistently observed only very small populations of DOPC:**4** liposomes (prior to light activation) by cryoTEM. Importantly, however, any potential membrane reorganization did not lead to disruption of liposome integrity and liposome encapsulated and membrane impermeable contents remained within the aqueous core of the liposome before, during and after light activation (Supplementary Fig. 6).

Following microinjection in the embryonic zebrafish (54–56 h post fertilization, hpf) and prior to light activation, DOPC:**4** liposomes (formulated at 1:1 molar ratio in all subsequent experiments) were freely circulating and did not significantly interact with RES cell types of the embryonic fish, namely blood resident macrophages and scavenging endothelial cells (SECs), at 1 hpi (Fig. 4d–f,h–j). Indeed, the exponential circulation lifetime decay of DOPC:**4** liposomes in the embryonic fish was very similar to that previously observed for 100 nm liposomes based

on the lipid composition of Myocet®[25]. Myocet® (lipid composition: POPC:cholesterol; 55:45) is a clinically approved liposomal-doxorubicin formulation designed to evade the RES, circulate freely and passively target solid tumors via the enhanced permeability and retention (EPR) effect[48]. In humans, the circulation half-life of Myocet® is 2.5 h[49]. In this case, like Myocet®, a significant fraction of photoactive DOPC:**4** liposomes remained in circulation >4 hpi (Fig. 4g and Supplementary Figs. 7 and 8 for individual images used for quantification). Upon in situ irradiation (15 min, 370 ± 7 nm, ~90 mW cm$^{-2}$, ~2.4 J per embryo, 1 hpi) of the entire zebrafish embryo, however, a dramatic change in liposome fate was observed, whereby DOPC:**4** → **3** liposomes were now visible as immobile punctae associated with all blood vessel walls (Fig. 4k–m) and largely removed from the circulating blood flow (Fig. 4g). The biodistribution of DOPC:**4** → **3** liposomes matched that of cationic DOPC:**3** (1:1) liposomes (Fig. 3b, c), confirming successful photoswitching of DOPC:**4** liposome surface charge —from (near) neutral to cationic—in situ and in vivo. In contrast, the biodistribution of freely circulating DOPC liposomes (100% DOPC content) was unaffected following identical irradiation procedures, confirming that the observed changes in biodistribution require the combination of both circulating DOPC:**4** liposomes as well as applied UV light (Supplementary Fig. 9). All UV irradiated embryos used in this study continued to develop normally without observable phenotypic abnormalities up to 6 dpf, confirming the suitability of this animal model for photoactivation studies[50]. Furthermore, any small potential increase in the temperature of the embryo as a result of UV irradiation will likely be counteracted by the remarkable resilience of the zebrafish embryo (from 1dpf) to heat stress[51].

Having shown photoswitching of liposome surface charge occurs within seconds (Fig. 4b), we next investigated the tissue level fate of i.v. administered DOPC:**4** liposomes, within the embryonic zebrafish, in real time (imaging rate: 1 frame per second, fps), before and during light-triggered surface charge switching. For this, a custom built, two-photon microscope was equipped with a 370 ± 7 nm LED, enabling alternating UV irradiation (95% UV duty cycle, Fig. 5a) and two-photon fluorescence imaging (see Supporting Information for setup, Supplementary Figure 10). For these experiments, we focused on a single plane (200 μm × 200 μm) of view which included both the dorsal aorta (DA) and posterior cardinal vein (PCV) to ensure potential liposome selectivity (venous vs. arterial endothelium) could be observed (Fig. 5b). From the acquired movie, two fundamental and competing interactions of cationic nanoparticles could be simultaneously observed, namely non-specific adsorption of liposomes to endothelial cells (ECs) and liposome aggregation in circulation (Supplementary Movie 1 and Fig. 5d for selected individual frames). Non-specific adsorption of liposomes to ECs was observed less than a minute after light activation, evident as emerging punctae of immobile fluorescence adhered to all blood vessel walls within the plane of view. The number of immobile punctae increased over time, and while there was no apparent selectivity for arterial or venous blood vessels, the largest number of liposomes were associated with the walls of the intersegmental vessel (ISV) connected to, and extending dorsally from, the PCV (Fig. 5d). ISVs are narrower blood vessels than both DA or caudal vein (CV) and the blood flow velocity within this vessel is reduced[52]. As a result, circulating cationic liposomes will spend an increased residence time within this vessel, compared to larger DA or CV blood vessels, under reduced shear stress[53,54]. This, in turn, presumably increases the propensity of cationic liposomes to adhere to the anionic, heparan sulfated endothelium of the ISV through direct electrostatic interactions.

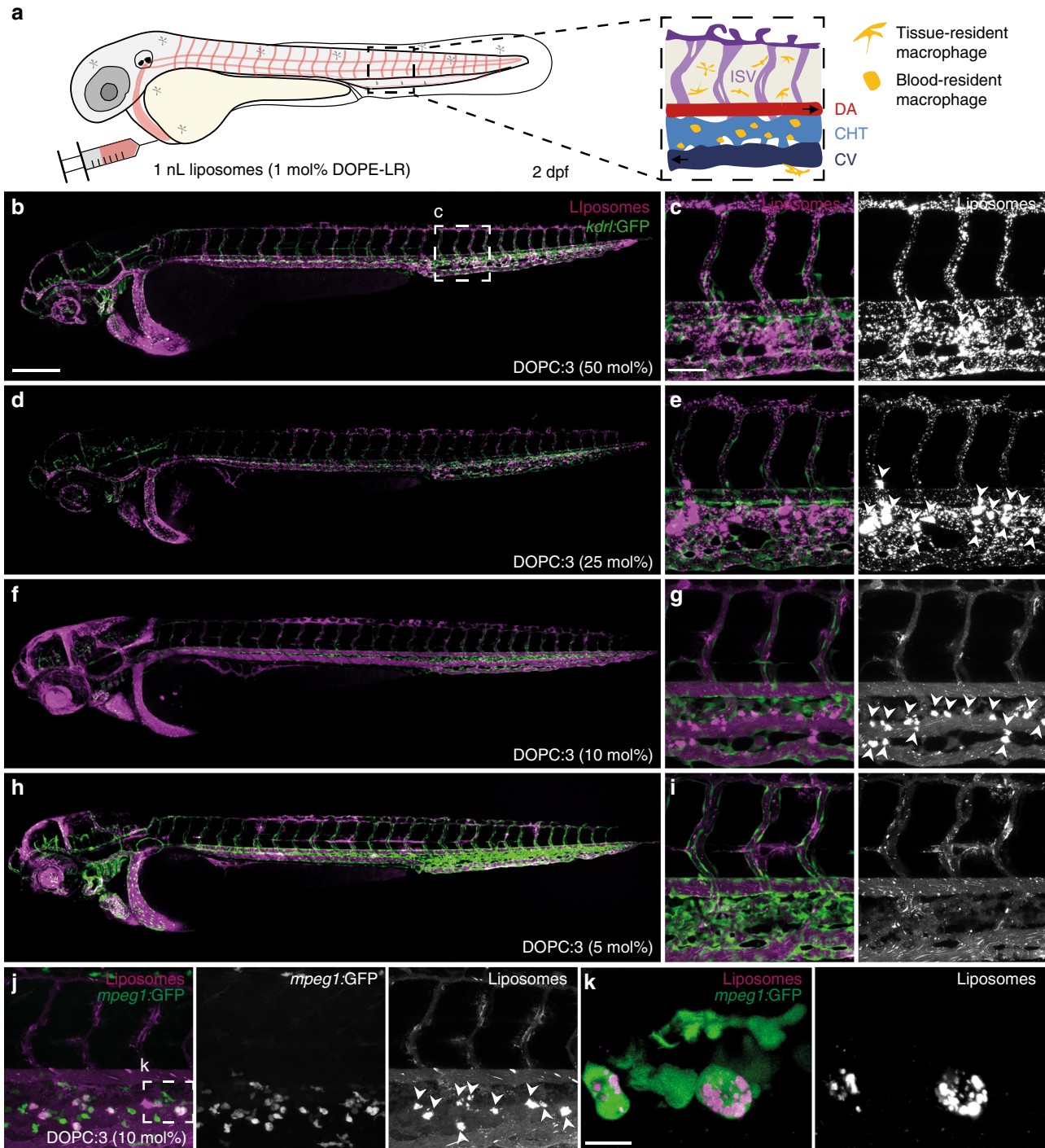

**Fig. 3 Biodistribution of cationic, DOPC:3 liposomes in embryonic zebrafish. a** Schematic showing the site of microinjection within a zebrafish embryo, two days post-fertilisation (dpf). Boxed region showing the organisation of blood vessels/macrophages within the tail of the embryo. DA dorsal aorta, CHT caudal hematopoietic tissue, CV caudal vein, ISV intersegmental vessel. Black arrows indicate direction of blood flow. **b–i** Biodistribution of DOPC liposomes containing cholesterylamine, **3**, at varying mol%. Whole embryo (×10 magnification) and tissue level (×40 magnification) views of liposome distribution in *kdrl:GFP* transgenic embryos, stably expressing GFP in all endothelial cells, at 1 hpi. White arrowheads indicate apparent liposome uptake within blood resident macrophages, based on location and cell morphology. **j**, **k** Tissue and cellular (×63 magnification) level views of DOPC:**3** (10 mol% **3**) liposome distribution in *mpeg1:GFP* transgenic embryos, stably expressing GFP in all macrophages, at 1hpi. Extensive fluorescence co-localization of liposomes and transgenic GFP confirmed the uptake of DOPC:**3** (10 mol% **3**) liposomes in blood resident macrophages of the zebrafish embryo. Slight variations in the positions of macrophages (between **j** and **k**) are due to macrophage migration during the time taken to change magnification settings on the confocal microscope. All liposomes **b–k** contained 1 mol% fluorescent lipid probe, DOPE-LR, for visualisation. Scale bars: 200 μm (whole embryo); 50 μm (tissue level); 10 μm (cellular level).

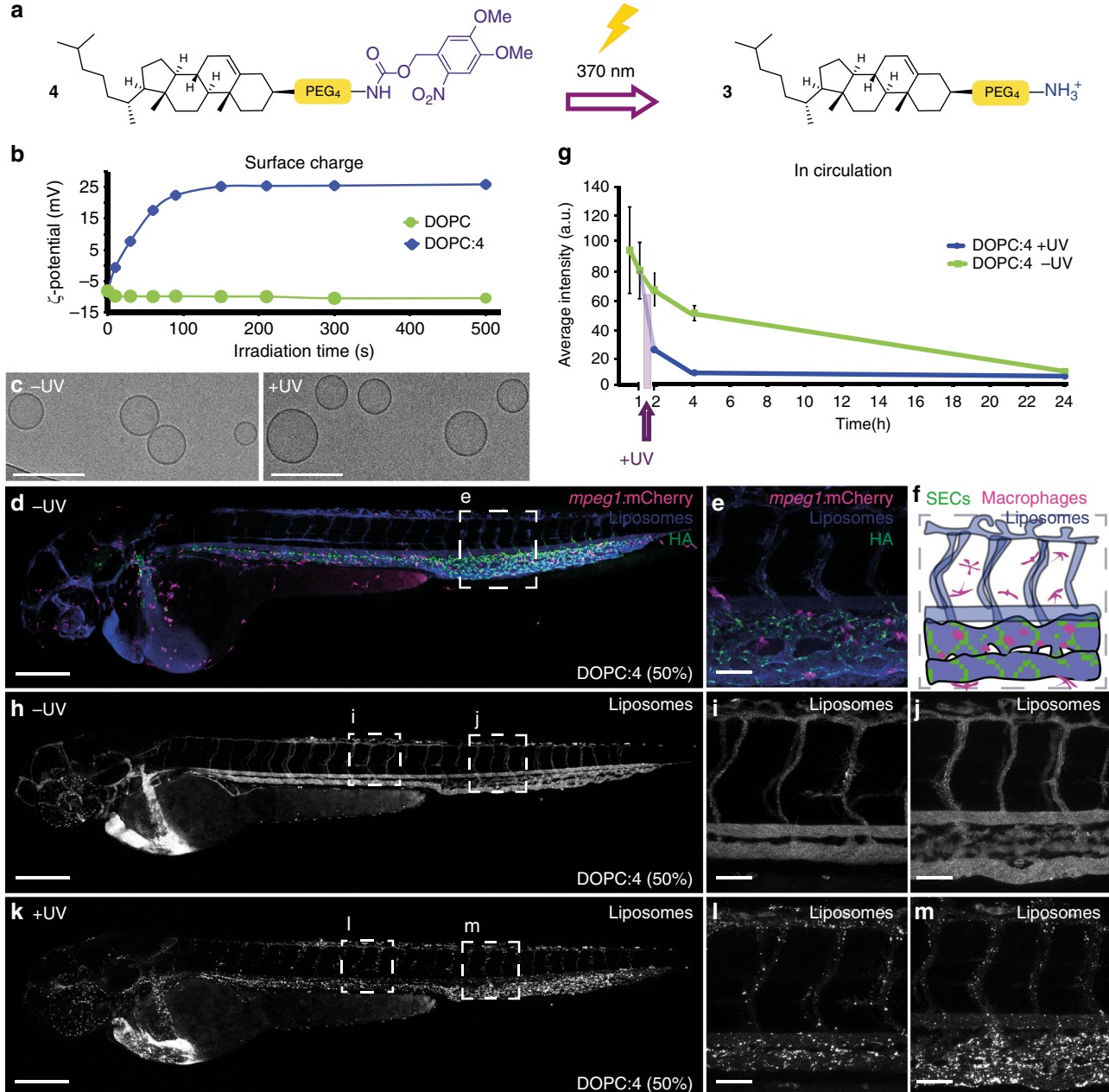

**Fig. 4 Photoswitching liposome surface charge in vitro and in vivo. a** Chemical structure of **4** and its photolysis to **3**. **b** Evolution of measured surface charge of DOPC:**4** liposomes (1:1) as a function of UV (370 ± 7 nm, 202 mW cm$^{-2}$) irradiation time. Note: batch-to-batch variation resulted in measured zeta potentials of DOPC:**4** → **3** liposomes ranging from +20 to +35 mV. Data presented is representative of liposomes used in Fig. 3h–m. 100% DOPC control liposomes demonstrate surface charge of liposomes without photoactive lipids is unaffected by UV irradiation. **c** Cryo-TEM images of DOPC:**4** before and after in situ irradiation (15 min, 370 ± 7 nm, 202 mW cm$^{-2}$). Scale bar: 200 nm. See Supplementary Information Fig. 5 for low magnification cryoTEM images. **d**, **e** Whole embryo and tissue level views of DOPC:**4** liposome biodistribution following co-injection with fluoHA in *mpeg1:mCherry* transgenic embryos (2 dpf). FluoHA is a specific in vivo marker of SECs and does not compete with liposome binding[25]. Liposomes (**d**, **e**) contained 1 mol% fluorescent lipid probe, DOPE-Atto633, for visualization. **f** Tissue level organization of macrophages and fluoHA-labelled SECs within the tail region of an *mpeg1:mCherry* embryo (2 dpf). **g** Quantification of DOPC:**4** liposome levels in circulation based on mean liposome fluorescence intensity in the lumen of the DA at 0.5, 1, 2, 4 and 24 hpi (measure of centre: median; error bars: standard deviation); $n = 6$ (0.5 and 1 hpi) and $n = 3$ (2, 4 and 24 hpi) individually injected embryos per formulation per time point (see Fig. S7 for individual images). **h–j** Whole embryo and tissue level views of DOPC:**4** liposome biodistribution in *kdrl:GFP* embryos, prior to UV irradiation, 1 hpi. **k–m** Whole embryo and tissue level views of DOPC: **4** → **3** liposome biodistribution in *kdrl: GFP* embryos, directly after in situ irradiation (15 min, 370 ± 7 nm, ~90 mW cm$^{-2}$, ~2.4 J per embryo), ~1.5 hpi. Liposomes (**h–m**) contained 1 mol% fluorescent lipid probe, DOPE-LR, for visualization. Scale bars (**d–m**): 200 μm (whole embryo); 50 μm (tissue level).

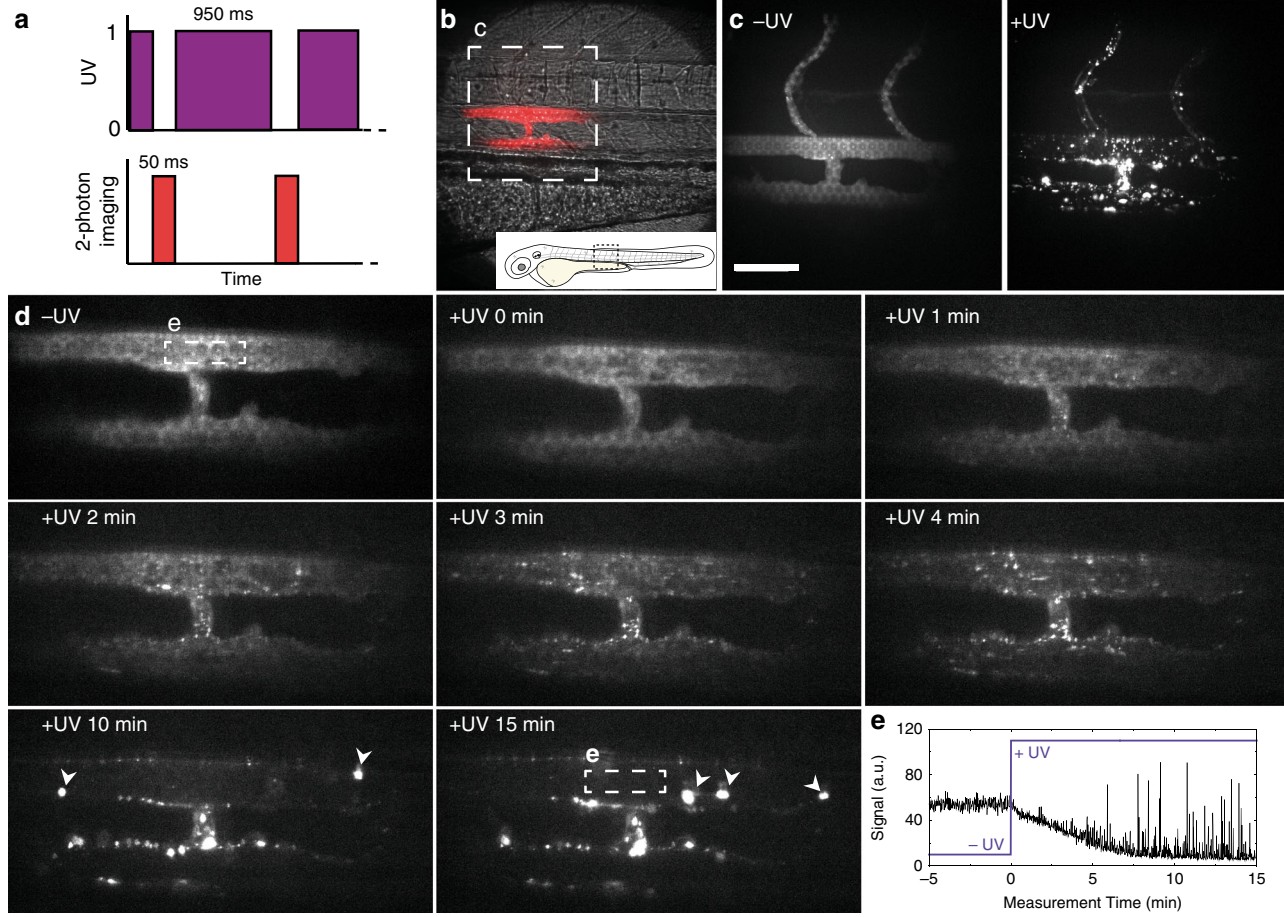

**Fig. 5 Real time multi-photon imaging of liposome surface charge switching in situ and in vivo. a** The zebrafish embryo was exposed to UV light between the acquisitions of the camera, resulting in a 95% duty cycle of UV illumination per frame. **b** Transmission image of the imaging location, emission image overlaid in red. **c** Maximum intensity projections of two-photon z-stacks (spanning the full width of the embryo) showing DOPC:**4** liposome distribution, before and after UV exposure. These images confirm the vessel connected to, and extending dorsally from the PCV, is an ISV. **d** Time-lapse images of DOPC:**4(→3)** liposome distribution before and during UV irradiation. In later timeframes, large clusters of liposomes (indicated with white arrowheads) were observed passing through the plane of view in circulation. **e** Mean fluorescence intensity within the ROI (lumen of the DA, white square in (**c**, 15 min). Liposome fluorescence intensity immediately decreased upon UV irradiation. High intensity spikes of fluorescence, due to large circulating liposome aggregates passing through the ROI, registered from 5 min after UV irradiation start. Liposomes contained 1 mol% fluorescent lipid probe, DOPE-LR, for visualisation. Scale bars: 50 μm.

Competing with the non-specific adsorption to the blood vessel walls, cationic liposome aggregation in circulation was also observed following light activation. This could be directly visualized as increasingly large and bright fluorescent particles passing through the plane of view in circulation (Supplementary Movie 1 and Fig. 5d, e). Aggregation of cationic liposomes is caused by the adsorption of anionic serum proteins/macromolecules to the newly revealed cationic nanoparticle surface[55,56], and we have recently shown cationic liposomes adsorb significantly more serum proteins than anionic or neutral liposomes in vitro[57]. Adsorption of this protein corona will not only mask underlying cationic surface charge (preventing direct electrostatic interaction with blood vessel walls) but will induce liposome aggregation and concomitant uptake in blood resident macrophages. Indeed, over the course of this research and in the absence of light activation, we have observed variable, low level uptake of DOPC:**4** liposomes within blood resident macrophages, predominantly within the CHT of the embryo. This may be due to incidental light exposure during experimental and microscopy procedures, partial photolysis and subsequent aggregation of liposomes in circulation, followed by irreversible recognition and clearance via the RES.

These simultaneous and competing interactions of cationic liposomes in vivo can be explained by the contrasting fates of DOPC:**4** → **3** liposomes as they transition through various intermediate charged states, from (near) neutral to cationic surface charge. In particular, during the light triggered transition of DOPC:**4** liposome surface charge, an intermediate physicochemical state, highly prone to blood-resident macrophage uptake (i.e. compositionally similar to DOPC liposomes containing 10 mol% **3**), is, at least momentarily, inevitable (Fig. 6a). The extent of macrophage uptake versus non-specific adsorption to ECs, should, therefore, be dependent on the residence time of partially activated DOPC:**4** → **3** liposomes in circulation. To test this hypothesis, we systematically reduced the applied UV light dose (from 75% to 10% UV duty cycle; 1.8–0.24 J per embryo, respectively) to extend the time taken for DOPC:**4** liposomes to transition from a (near) neutral to cationic surface charge in situ and in vivo (Fig. 6b–i). In this way, the biodistribution of DOPC:**3** liposomes, containing varying mol% **3** (Fig. 3b–i), could be replicated. Most striking, at 25% applied light (0.6 J per embryo), DOPC:**4** → **3** liposomes were predominantly taken up by blood resident macrophages within the CHT of the embryonic zebrafish, analogous to (near)

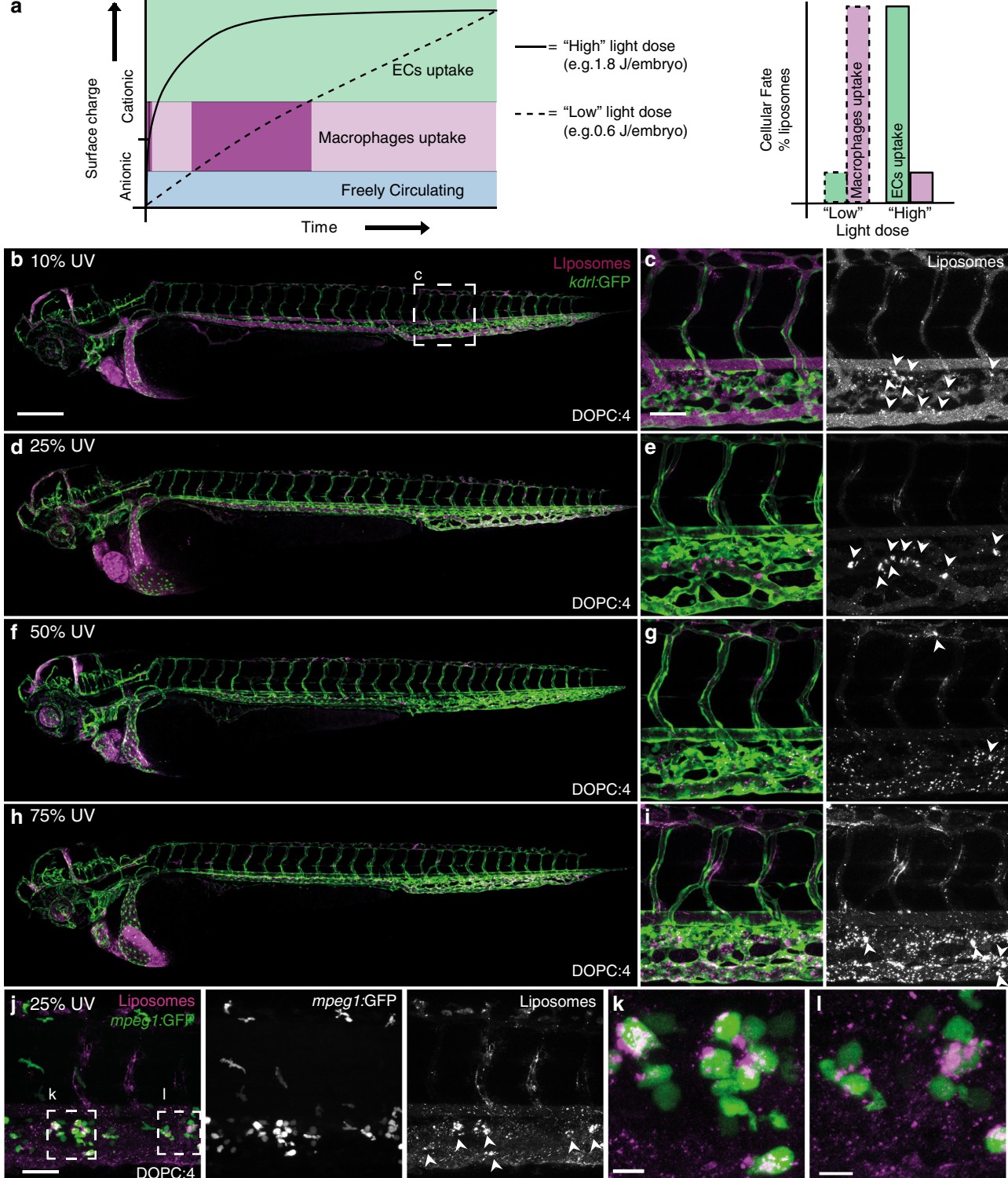

**Fig. 6 Effect of light dose on the in vivo fate of DOPC:4 liposomes. a** Reducing the UV light dose applied to the embryo increases the time taken for DOPC:**4** liposomes to transition to DOPC:**3** liposomes. This, in turn, increases the resident time spent by DOPC:**4** → **3** liposomes at an intermediate cationic surface charge density leading to irreversible clearance of liposomes by blood resident macrophages. **b-i** Whole embryo and tissue level views of DOPC:**4** liposome biodistribution in *kdrl:GFP* embryos following embryo irradiation with variable light doses (15 min, 370 ± 7 nm, ~90 mW cm$^{-2}$, UV duty cycle stated for each image). Apparent liposome uptake in blood resident macrophages highlighted with white arrowheads. **j-l** Tissue level and zoomed views of DOPC:**4** → **3** liposome biodistribution in *mpeg1:GFP* embryos following embryo irradiation at 25% UV duty cycle (15 min, 370 ± 7 nm, 0.6 J per embryo). Liposomes contained 1 mol% fluorescent lipid probe, DOPE-LR, for visualisation. Scale bars: 200 μm (whole embryo); 50 μm (tissue level), 10 μm (zoomed).

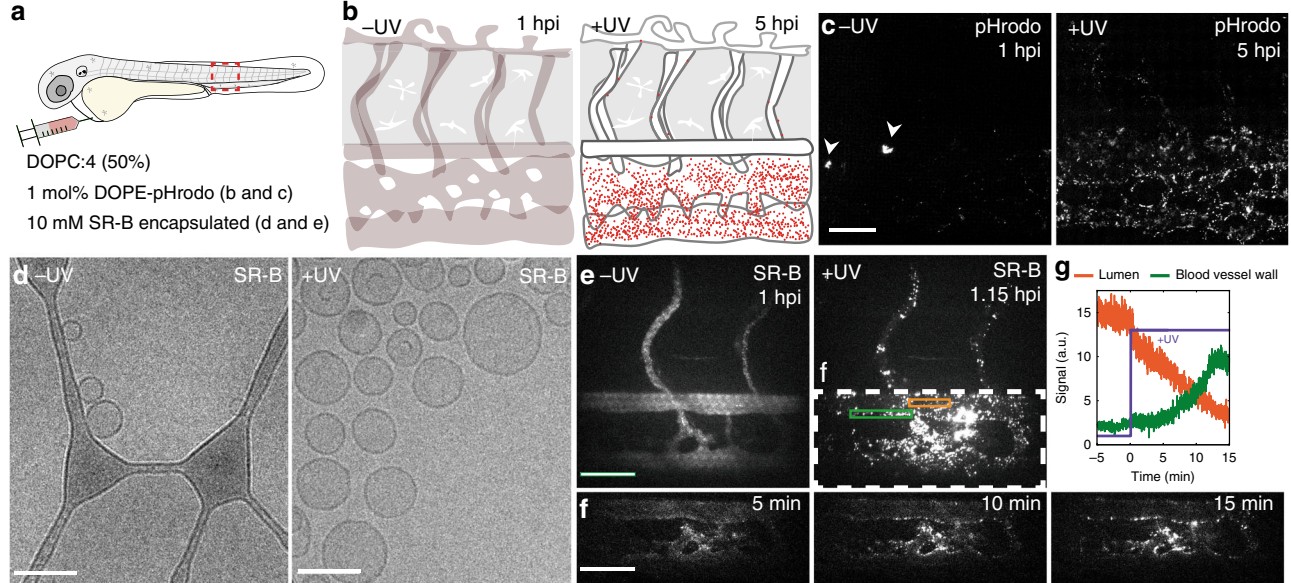

**Fig. 7 Cellular fate of DOPC:4 → 3 liposomes and their encapsulated payloads. a, b** Schematics showing the site of microinjection within a 2 dpf embryo and the evolving fluorescence of pHrodo-containing DOPC:**4 → 3** liposomes within the tail region of a wildtype (AB/TL) embryonic zebrafish, over time. The fluorescence intensity of pHrodo increases >100-fold in mildly acidic environments (e.g. late endosomes/lysosomes, pH < 6) and is, therefore, particularly apparent within SECs—cells with exceptionally high lysosomal activity. **c** Tissue level views of evolving pHrodo-associated fluorescence over time either in the absence of UV irradiation and following in situ UV irradiation (15 min, 370 ± 7 nm, 2.4 J cm$^{-2}$). In the absence of UV irradiation, pHrodo-associated fluorescence is observed within a small number of cells within the CHT of the embryo (white arrowheads). Liposomes contained 1 mol% DOPE-pHrodo for visualisation. Scale bar: 50 μm. **d** CryoTEM images of SR-B filled, DOPC:**4** liposomes before and after in situ irradiation (15 min, 370 ± 7 nm, 202 mW cm$^{-2}$). Scale bars: 100 nm. **e** Maximum intensity projections of two-photon z-stacks (spanning the full width of the embryo) showing SR-B filled DOPC:**4** liposome distribution, before and after UV irradiation. Scale bar: 50 μm. **f** Time-lapse images of SR-B filled DOPC: **4 → 3** liposome distribution during UV irradiation. Scale bar: 50 μm. **g** Mean SR-B fluorescence intensity within ROIs (lumen of the DA, orange line; DA vessel wall, green line, shown in **e**), before and during UV irradiation. SR-B fluorescence intensity in circulation decreases upon UV irradiation with a concomitant increase in SR-B fluorescence intensity associated with the DA blood vessel wall. Liposomes (**e–g**) containing encapsulated SR-B (10 mM) and otherwise unlabelled.

neutral DOPC:**3** liposomes (10 mol% **3**) (Fig. 6d, e, j–l). This result confirmed that an extended residence time of only partially activated DOPC:**4 → 3** liposomes leads to irreversible clearance of liposomes by the RES.

**Light-triggered liposomal cell uptake and payload delivery.** Next, we investigated the intracellular fate of DOPC:**4** liposomes following in situ and in vivo surface charge switching. To verify endocytosis of DOPC:**4 → 3** liposomes, a pH-sensitive fluorescent lipid probe (DOPE-pHrodo, 1 mol%) was incorporated within the DOPC:**4** liposome membrane. The fluorescence intensity of pHrodo increases >100-fold in mildly acidic environments (e.g. late endosomes/lysosomes, pH < 6). To accurately assess evolving pHrodo fluorescence, wild type (AB/TL) embryos were used to avoid potential fluorescence bleed through from transgenic fluorescent markers. Following i.v. injection and in the absence of light, pHrodo-associated fluorescence was observed in a small number of cells within the CHT of the embryonic fish (Fig. 7a, b). While the absence of cell specific (transgenic) fluorescent markers does not allow for definitive identification of this cell type, its location within the CHT and rounded morphology (delineated by pHrodo fluorescence), is characteristic of low-level phagocytic uptake of DOPC:**4** (→**3**) liposomes within blood resident macrophages, as previously mentioned. Following in situ light irradiation, however, increasing pHrodo fluorescence, primarily associated with SECs, was observed over time (up to 5 hpi) (Fig. 7c). This timeframe suggests a significant fraction of DOPC:**4 → 3** liposomes are not only endocytosed by SECs but remain within endosomes during trafficking and maturation to late endosomes/lysosomes. This would be consistent with the very high lysosomal activity of SECs, whose primary

physiological role in the body is to recognize, clear and breakdown endogenous and pathogenic waste from the blood[58]. It is also possible that a fraction of endocytosed, cationic DOPC:**4 → 3** liposomes, within SECs, macrophages or other ECs, manage to successfully escape endosomes prior to the first imaging time point. Within the cytosol and beyond the acidic endosome micro-environment, potential intracellular fluorescence associated with pHrodo probes would not be visible.

Having confirmed endocytosis of light-activated DOPC:**4 → 3** liposomes in vivo, we finally investigated the in vivo fate of liposome encapsulated and membrane impermeable payloads, following light triggered surface charge switching. For this, self-quenching concentrations of water soluble, sulforhodamine B (SR-B, 10 mM) were passively encapsulated within DOPC:**4** liposomes. As for empty liposomes, the morphology and size of the SR-B filled DOPC:**4** liposomes did not significantly change following irradiation (Fig. 7d). To monitor the fate of liposome encapsulated SR-B in vivo, we again performed alternating UV irradiation and two-photon fluorescence imaging (1fps) within live embryos (Fig. 7e–g and Supplementary Movie 2). In this case, we focused on a single plane (200 μm × 200 μm) of view to include the DA, CHT and CV. This region of the embryo includes a significant population of SECs, as well as blood resident macrophages, within which DOPC:**4 → 3** liposomes extensively accumulate. From the acquired movies, SR-B filled DOPC:**4** liposomes, prior to light activation, appeared freely circulating, evident as homogenous, low level (quenched) SR-B fluorescence, confined within the zebrafish vasculature. Upon light triggered activation (95% UV duty cycle, 2.3 J per embryo) and surface charge switching, however, localized and de-quenched SR-B could be seen as increasingly bright, fluorescent punctae,

associated with all blood vessel walls but most prevalent throughout the sinusoid-like network of CHT blood vessels. Again, it is likely that DOPC: $4 \rightarrow 3$ liposome association and uptake within the CHT is enhanced due to the reduced blood flow velocities and sheer stresses experienced by liposomes within this sinusoidal-like tissue. Likewise, the co-existence of immobile (arrows, Supplementary Movie 2) as well as highly mobile (arrowheads, Supplementary Movie 2) SR-B fluorescent punctae is again indicative of intracellular SR-B delivery to both ECs (immobile) and blood resident macrophages (mobile). As a representative membrane impermeable, small molecule cargo, the successful intracellular delivery of liposome-encapsulated SR-B serves as promising evidence that this proof-of-concept technology will likely be transferable to other, therapeutically relevant (and membrane impermeable) payloads.

## Discussion

The discovery of simple and effective targeted drug delivery systems should be preceded by a thorough understanding of the nano-bio interactions involved[59]. Here, we exploit the well characterised and contrasting fates of differently charged nanoparticles in vivo, however, our ability to rationally design a system that is both simple and effective has relied on comprehensive in vivo interrogation of all aspects of this technology (i.e. different light dosages, surface charge densities, interactions with the RES). To this end, the embryonic zebrafish has provided an invaluable pre-clinical in vivo screening platform, offering unprecedented opportunities to assess, analyse and optimise nanoparticle behaviour over an entire live organism (i.e. visualising total injected doses), at cellular resolution and in real time[42,60,61]. Furthermore, the presence and conserved function of key RES cell types enables predictive assessment of fundamentally important in vivo clearance mechanisms of nanoparticles[25,43]. It is important to stress here, however, that these predictions are strictly qualitative. Given the significant differences in relative numbers of RES cells (i.e. SECs vs blood resident macrophages), RES tissue size and organization, quantitative predictions (based on observations in the embryonic zebrafish) of nanoparticle clearance by the mammalian RES are not yet possible. As such, the embryonic zebrafish should not replace experiments in larger (mammalian) models but, instead, should be used to guide and optimize nanoparticle design prior to first injections in higher vertebrates.

At a fundamental level, the ability to visualise the formation of a cationic nanoparticle in situ has revealed, for the first time, the co-existence of two competing interactions of cationic nanoparticles occurring simultaneously in vivo, namely non-specific adsorption to blood vessel walls and opsonisation in circulation. This observation not only highlights the importance of considering the fate of all intermediate physicochemical states of stimuli-responsive nanoparticles as they transition from A to B, but, more generally, suggests that the fate of any given cationic nanoparticle is dependent on its surface charge density. Our data indicates that above a certain cationic surface charge threshold, i.v. administered nanoparticles will predominantly stick to (and be internalised by) endothelial cells, particularly in blood vessels with reduced blood flow velocity, while below this charge threshold, nanoparticles will tend to aggregate in circulation and be subsequently cleared by the RES. In our experience, i.v. administered liposomes with a measured surface charge >20 mV will tend to stick to endothelial cells, whereas those with a surface charge between +5 and +20 mV will tend to aggregate in circulation. Importantly, while we believe a threshold value will apply to all nanoparticle classes, this will likely vary depending on the surface chemistry and self-assembly of any individual nanoparticle and should be determined on a case-by-case basis.

In terms of targeting, the exclusive use of light as trigger forgoes any requirement for exploitable differences between target and non-target tissues (e.g. passive targeting of solid tumours via the EPR effect). As such, this liposome technology has the potential to be transferable to any light accessible tissue. Given the poor tissue penetration and significant potential cytotoxicity of using short wavelength UV-A light[62,63], however, we would aim future efforts at replacing o-nitrobenzyl chemistries with photocages sensitive to longer wavelength light or two-photon activation. In this vein, a family of zwitterionic BODIPY-derived photocages have recently been reported that can be efficiently cleaved using single photon visible or NIR light[39]. In theory, these photocages, connected to a cholesterylamine lipid anchor and incorporated within DOPC liposomes should not affect the surface charge and, therefore, the biodistribution of photoactive liposomes. In addition, light can be focused with precise spatial resolution. This has been exemplified by the clinical application of Visudyne®—a liposome-photosensitizer (verteprofin) formulation, administered intravenously and indicated for the photodynamic therapy of age-related macular degeneration (AMD)[64]. In this case, non-thermal, red light (689 nm) is applied to the eye of a patient to trigger localised therapy. Unfortunately, given the small size (2–3 mm in length) of the zebrafish embryo and the practical difficulties in ensuring no incident or scattered UV-A light reached the dark side of the agar embedded embryo, we have been unable to demonstrate localised liposome surface charge switching and intracellular uptake within the embryonic fish in this study.

In conclusion, we describe a liposome technology that successfully couples complete external control of in vivo liposome targeting together with the transport of encapsulated and membrane impermeable cargos across cell membranes. While these combined features are unique in the context of stimuli-responsive drug delivery systems (reviewed in refs. [65,66]), including those for which charge switching is central to function (reviewed in ref. [67]), the stand out feature of these liposomes is undoubtedly their compositional simplicity. The last decades have seen the empirical design of increasingly more complex nanomedicine designs, but it is now generally acknowledged that this approach has impeded rather than promoted the clinical translation of new nanomedicines[59,68–70]. In contrast, clinically approved and targeted nanomedicines tend to be compositionally simple[49], with designs based on robust physicochemical principles (e.g. PEGylation to improve circulation lifetimes)[71] and well characterised and exploitable, albeit now clinically questionable[72,73], biological phenomena (e.g. the EPR effect of select solid tumours)[74]. Following these principles, we have designed a simple and effective proof-of-concept liposome technology, composed of just two lipids, based on, and preceded by, a thorough understanding of both the physicochemical and in vivo nano-bio interactions involved. As such, it is our hope that this study, and in particular the tools and methods employed, will expedite a transition from the empirical design of increasingly complex nanomedicines to the rational design of new, simple and effective nanomedicines[75].

## Methods

**Materials and reagents**. 1,2-dioleoyl-*sn*-glycero-3-phosphocholine (DOPC), 1,2-dioleoyl-3-trimethylammonium-propane (DOTAP), 1,2-dioleoyl-*sn*-glycero-3-phosphoethanolamine-N-(lissamine rhodamine B sulfonyl) (Rhodamine-PE) were purchased from Avanti Polar Lipids (Alabaster, AL, US). 1,2-Dioleoyl-*sn*-glycero-3-phosphoethanolamine-Atto 633 was purchased from ATTO-TEC GmbH (Germany). Additional DOPC was purchased from Lipoid GmbH. Cholesterol and sulforhodamine B (SR-B, sodium salt) were purchased from Sigma Aldrich. pHrodo™ Red, succinimidyl ester (pHrodo™ Red, SE) was purchased from Thermo Fisher Scientific. DOPE-pHrodo was prepared through conjugation of DOPE with pHrodo succinimidyl ester under basic conditions, followed by column chromatography[76]. Fluorescein-labeled hyaluronic acid (fluoHA) was prepared through conjugation of hyaluronic acid (100 kDa) with fluorescein isothiocyanate (Isomer I,

Sigma-Aldrich) under mildly basic conditions, followed by ethanol precipitation[77]. All other chemical reagents were purchased at the highest grade available from Sigma Aldrich and used without further purification. All solvents were purchased from Biosolve Ltd. For anhydrous reactions, solvents were dried over activated molecular sieves (3 Å, 4–8 mesh). *HEPES buffer*: HEPES (10 mM) adjusted to pH 7.4 with 1 M aqueous NaOH. Ultrapure MilliQ® water, purified by a MilliQ Advantage A10 water purification system from MilliPore, was used throughout.

**Chemical synthesis and characterization**. TLC analysis was performed using aluminium TLC plates, coated with 0.25 mm silica gel 60 $F_{254}$ from Merck KGaA. Plates were visualized by UV absorption at 254 nm and/or staining with $KMnO_4$ solution. Flash-column chromatography was performed using silica gel 60 (particle size of 40-63 μm) from Merck KGaA. $^1H$ and $^{13}C$ NMR spectra were acquired using an Avance DPX-300MHz or AV-400 MHz NMR spectrometer from Bruker at room temperature. Chemical shifts are given in ppm with tetramethylsilane (TMS) or residual solvent ($CDCl_3$: 7.26 ppm for $^1H$ NMR and 77.2 ppm for $^{13}C$ NMR) as internal standard. Signal multiplicity is described with common abbreviations: singlet (s), broad singlet ($s_{br}$), doublet (d), triplet (t), quartet (q), multiplet (m). Coupling constants are given in Hz. See Supplementary Information for detailed chemical synthesis protocols and Supplementary Figs. 12–18 for all $^1H$ NMR spectra. High-resolution mass spectrometry (HRMS) were recorded on a Thermo Scientific LTQ Orbitrap XL. UV absorption spectra were measured using a Cary 3 Bio UV-vis spectrometer (Cary WinUV software, version 3.0, Agilent), scanning from 200 to 550 nm at 1 nm intervals. Scan rate: 120 nm min$^{-1}$.

**Light source and actinometry**. A commercially available 375-nm LED (Maximum measured wavelength = 370 nm, FWHM = 13.4 nm; H2A1-H375-S, Roithner Lasertechnik, Vienna, Austria), driven by a custom-built LED driver ($I = 350$ mA), was used as UV light source in all cases. The optical power density of the LED light source was determined using an integrating sphere setup. For this, the LED was positioned precisely 5 cm above the 6.0 mm aperture of an integrating sphere (AvaSphere-30-IRRAD, Avantes, Apeldoorn, The Netherlands). This sphere was connected by an optical fiber (FC-UV600-2, Avantes) to a UV-Vis spectrometer (AvaSpec-ULS2048L StarLine CCD spectrometer, Avantes). The setup was calibrated using a NIST-traceable calibration light source (Avalight-HAL-CAL-ISP30, Avantes). The LED was switched on, and allowed to warm up for 1 min, before a spectrum was recorded (see Supplementary Fig. 11 for measured UV-Vis spectrum). The obtained spectrum was integrated to obtain the total incident optical power density (in mW cm$^{-2}$). Light dosages (J per embryo) were obtained by multiplying the optical power density by the irradiation time. Average embryo surface area used was 0.03 cm$^2$ (0.1 × 0.3 cm). Precise irradiation setups are detailed within experimental descriptions.

**Liposome preparation**. All liposomes (without encapsulated payloads) were formulated in either (deionized) $H_2O$ or 10 mM HEPES buffer at a total lipid concentration of 4 mM. Individual lipids, as stock solutions (1–10 mM) in chloroform, were combined to the desired molar ratios and dried to a film, first under a stream of $N_2$ then >1 h under vacuum. Large unilamellar vesicles were formed through extrusion above the $T_m$ of all lipids (room temperature, Mini-extruder, Avanti Polar Lipids, Alabaster, US). Hydrated lipids were passed 11 times through 2 × 400 nm polycarbonate (PC) membranes (Nucleopore Track-Etch membranes, Whatman), followed by 11 times through 2 × 100 nm PC pores. All liposome dispersions were stored at 4 °C. All liposomes were stable for at least 1 month (in the dark).

**Size and zeta potential measurements**. Particle size and zeta potentials were measured using a Malvern Zetasizer Nano ZS (software version 7.13, Malvern Panalytical). For DLS (operating wavelength = 633 nm), measurements were carried out at room temperature in water or HEPES (10 mM) buffer at a total lipid concentration of ~100 μM. Zeta potentials were measured at 500 μM total lipid concentration, using a dip-cell electrode (Malvern), at room temperature. For liposomes formulated in water, aq. NaCl was added to the liposome solution to a final concentration of 10 mM NaCl before zeta potential measurement. All reported DLS measurements and zeta potentials are the average of three measurements. For DLS and zeta potential experiments monitoring changes following light activation, liposomes were irradiated (370 ± 7 nm, 202 mW cm$^{-2}$) in quartz cuvettes with the LED mounted at a distance of 1 cm from the sample. The same liposome sample was used for time course DLS and zeta potential measurements.

**Cryogenic transmission electron microscopy**. Liposomes (3–6 μL, 4 mM total lipid concentration) were applied to a freshly glow-discharged carbon 200 mesh Cu grid (Lacey carbon film, Electron Microscopy Sciences, Aurion, Wageningen, The Netherlands). Grids were blotted for 1, 2 or 3 s at 99% humidity in a Vitrobot plunge-freezer (FEI Vitrobot™ Mark III, Thermo Fisher Scientific). Cryo-EM images were collected on a Talos L120C (NeCEN, Leiden University) operating at 120 kV. Images were recorded manually at a nominal magnification of ×17,500 or ×36,000 yielding a pixel size at the specimen of 5.88 or 2.90 ångström (Å), respectively. For cryoTEM images monitoring changes following light activation, liposomes were irradiated (15 mins, 370 ± 7 nm, 202 mW cm$^{-2}$) in quartz cuvettes

with the LED mounted at a distance of 1 cm from the sample. The same liposome sample was used for before and after UV.

**Sulforhodamine-B encapsulation and characterization**. DOPC:**4** (1:1) lipid films (10 mM total lipids) were hydrated with HEPES buffer (1 mL) containing Sulforhodamine-B (SR-B) (10 mM) and formulated by extrusion, as described for empty liposomes. Un-encapsulated SR-B was removed by size exclusion chromatography (illustra™ NAP™ Sephadex™ G-25 DNA grade pre-made columns (GE Healthcare)) following the supplier's instructions. Eluted liposomes with encapsulated SR-B were diluted 2.5x during SEC (to ~4 mM [total lipid]) and injected without further dilution.

*Contents leakage assay*. SR-B leakage from DOPC:**4** (1:1) liposomes, before and after light activation, was monitored using a TECAN Infinite M1000 Fluorescence Plate Reader and were performed in 96-well plates (PP Microplate, solid F-bottom (flat), chimney well) at room temperature. Final experimental volume in each well was 200 μL. To monitor SR-B leakage (and dye de-quenching) during photo-activation, fluorescence emission (excitation: 520 nm; emission: 580 nm) was measured every 20 s for 600 s, the sample was then irradiated (20 mins, 370 ± 7 nm, 202 mW cm$^{-2}$) in a quartz cuvette, with the LED mounted at a distance of 1 cm from the sample, returned to the 96-well plate and fluorescence emission measured for a further 10 mins. After this, Triton X-100 (10 μL, 1% w/v) was added to the sample well (10 s agitation) to solubilize liposomes and release any remaining encapsulated SR-B.

**Zebrafish husbandry, injections and irradiation setup**. Zebrafish (*Danio rerio*, strain AB/TL) were maintained and handled in accordance with guidelines from the European Convention on the protection of vertebrate animals used for experimental and other scientific purposes[78], and in compliance with the directives of the local animal welfare committee of Leiden University. Fertilization was performed by natural spawning at the beginning of the light period, and eggs were raised at 28.5 °C in egg water (60 μg mL$^{-1}$ Instant Ocean sea salts). The following previously established zebrafish lines were used: *Tg(kdrl:eGFP)*[s84][79], *Tg(mpeg1: GFP)*[gl22][80], *Tg(mpeg1:mCherry)*[gl23][80]. Liposomes were injected into 2-day old zebrafish embryos (52-56 hpf) using a modified microangraphy protocol[81]. Embryos were anesthetized in 0.01% tricaine and embedded in 0.4% agarose containing tricaine before injection. To improve reproducibility of micro-angiography experiments, 1 nL sample volumes were calibrated and injected into the sinus venous/duct of Cuvier. A small injection space was created by penetrating the skin with the injection needle and gently pulling the needle back, thereby creating a small pyramidal space in which the liposomes were injected. Successfully injected embryos were identified through the backward translocation of venous erythrocytes and the absence of damage to the yolk ball. For embryo irradiation, the UV source (370 ± 7 nm) was positioned approximately 1.5 cm above the agar-embedded embryo (~90 mW cm$^{-2}$). 15 min total irradiation time (~2.4 J per embryo light dose) was used in all cases of embryo irradiation followed by confocal imaging. For experiments monitoring changes in liposome biodistribution following light triggered surface charge switching, the same embryo was imaged before and after UV irradiation.

**Confocal image acquisition and quantification**. Zebrafish embryos were selected according to successful injections and randomly picked from a dish of 20–60 successfully injected embryos. At least four zebrafish were visualized and the most representative zebrafish was imaged by a Leica TCS SPE or SP8 confocal microscope (Leica Application Suite X software, version 3.5.5.19976, Leica Microsystems). Confocal z-stacks were captured using a 10x air objective (HCX PL FLUOTAR), a ×40 water-immersion objective (HCX APO L) or 63x water-immersion objective (HC PL APO CS). For whole-embryo views, 3 overlapping z-stacks were captured to cover the complete embryo. Laser intensity, gain and offset settings were identical between stacks and experiments. Images were processed and quantified using the Fiji distribution of ImageJ[82,83]. For quantification of liposome circulation lifetime decay, at least three individual embryos (biological replicates) were imaged using confocal microscopy at each time point. Quantification (not blinded) was performed on ×40 confocal z-stacks (optical thickness of 2–3 μm per slice) using methods and ImageJ macros previously described[25]. Median values are reported.

**Two-photon setup and image acquisition**. A custom-built two-photon multifocal microscope was used for simultaneous UV irradiation and two-photon fluorescent imaging (see Supplementary Fig. 10 for schematic of the multiphoton microscope setup). A femtosecond pulsed Ti:Sa laser set at 830 nm (Coherent, Chameleon Ultra) was used as excitation source. Multifocal illumination of the sample was achieved by a diffractive optical element (DOE, custom made by Holoeye) which splits the laser beam into an array of 25 ×25 foci. A virtual light sheet was created by spiral scanning the foci within the 50 ms exposure time of the camera using a fast-scanning mirror (Newport, FSM-300-1)[84]. The virtual light sheet was focused and emission photons collected by a 25×, high-NA water-dipping objective (Nikon, CFI75 Apochromat 25XC W). The objective was positioned onto a piezo stage (P-726 PIFOC, PI) for z-stack measurements. Emission light was separated from the

excitation path by a dichroic mirror (700 dcxr, Chroma). After passing through a 700 nm short pass filter, emission photons were detected with a 2048 × 2048 pixel sCMOS camera (Hamamatsu, Orca Flash 4.0 V2). Emission images were taken at the start of the experiment with a white LED. After the emission images, the UV-LED was installed at the location of the white LED. The UV LED was positioned ~1.5 cm (~90 mW cm$^{-2}$) above the sample and on/off-times were timed by the same data acquisition card (USB-6226, National Instruments) which triggered the camera. Simultaneous UV irradiation and two-photon fluorescent imaging was performed 1 h post injection. To ensure stability, embryos were imaged for 15 min before the measurement. Once embryos were stable, images were taken every 1 s for 5 min. After 5 min the UV lamp was turned on and switched off only during camera exposure. Two-photon microscopy data was processed using custom-built LabVIEW software (version 2018 SP2, National Instruments).

**Statistics and reproducibility**. All experiments presented in the main manuscript were repeated at least twice with the exception of Fig. 6b,c. All replicate experiments were performed using freshly prepared liposomes. Unless clearly stated in the manuscript text (e.g. varying macrophage uptake prior to UV activation), all replicate experiments were successful and confirm the presented data. All experiments presented in Supplementary Information were repeated at least twice, with the exception of Supplementary Figs. 1 and 9. All replicate experiments were performed using freshly prepared liposomes. For all experiments performed in embryonic zebrafish, at least four embryos were randomly selected (from a pool of >20 successfully injected embryos) and imaged (low resolution microscopy). Unless clearly stated in the manuscript text (e.g. varying macrophage uptake prior to UV activation), all imaged embryos showed consistent results and confirmed the presented data. From these four embryos, one was selected for high resolution, confocal microscopy. No statistical analysis is performed in this work.

**Reporting summary**. Further information on research design is available in the Nature Research Reporting Summary linked to this article.

## Data availability
Data supporting the findings of this paper are available from the corresponding authors upon reasonable request. Source data (raw confocal z-stacks and collated data as single Excel sheet) underpinning the data presented in Fig. 4g have been deposited within the public image database, figshare.com (https://doi.org/10.6084/m9.figshare.12387629). Source data are provided with this paper.

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

## Acknowledgements

Many thanks to Felipe Pozetti de Melo Faria (Erasmus student), Myrthe Crombaghs (undergraduate student) and Mitchell Lansink (undergraduate student) who worked on early iterations of this idea that, although unsuccessful, invaluably helped to shape the research presented here. This work benefited from access to the Netherlands Centre for Electron Nanoscopy (NeCEN) at Leiden University, an Instruct-ERIC centre, with technical assistance from Ludovic Renault, Christophe Dievolder, Rebecca Dillard and Jamie Depelteau. This work was supported financially by the Netherlands Organization for Scientific Research (NWO-VICI—project no. 724.014.001 (awarded to A.K.)—supporting F.C., G.A.; NWO-VICI (awarded to J.v.N)—project no. 680.47.616—supporting R.V.), a M-ERA grant (awarded to A.K)—supporting F.C.; a Leiden/Huygens prize scholarship (P.P.), a Chinese Scholarship Council grant (L.K.), and the Biomolecular Nanoscale Engineering Center (BioNEC), a Centre of Excellence funded by The Villum Foundation (grant no. VKR022710; S.V and A.R.).

## Author contributions

F.C. conceived the research. F.C., G.A., R.V. and S.B. designed the experiments. G.A, L.K., R.V., A.R, P.P. and F.C. carried out the experiments. G.A., L.K., R.V., A.R., P.P., M.S.M., S.B., S.V., J.v.N., A.K. and F.C. analysed the data. M.S.M. performed the light actinometry. R.V. and J.v.N. designed and built the multiphoton microscope. F.C., G.A., R.V. wrote the paper with feedback from all authors. A.K. supervised the research.

## Competing interests

The authors declare no competing interests.
