## [Peer Review File · Nature Communications]

Reviewers' Comments:

Reviewer #1:

Remarks to the Author:

The manuscript entitled "Light-triggered switching of liposome surface charge directs the intracellular delivery of membrane impermeable payloads in vivo" by G. Arias-Alpizara et al. describes light-induced switching of surface charges of liposomal carriers to induce cellular uptake of long-circulating membrane vesicles. Proof-of-concept is provided using a zebrafish embryo in vivo model. This novel concept is very interesting in that surface charge switching using photo-activated lipids can be used to change the pharmacokinetic properties of circulating nanoparticles within the living organism.

Major comments:

line 194: Content leakage of liposomes was monitored over a time period of 30 minutes (Figure S6). What about stability over prolonged periods of time (e.g. storage at low temperature) of non-irradiated samples or irradiated samples during 24 hours?

line 217: "Having previously shown photoswitching of liposome surface charge occurred within seconds": please provide the reference

Liposome aggregation in vivo due to changes in the protein corona before or after irradiation will trigger cellular uptake by macrophages. Were DLS experiments carried out in presence as well as absence of plasma proteins?

Please explain the advantages and limitations of the used zebrafish model in more detail. Is it possible to extrapolate from data obtained in zebrafish to higher vertebrates?

Minor comments:

Line 85: "In our experience, anionic nanoparticles...": please provide the reference or does this relate to ref. 25?

line 98: "In the contrasting fates..." should read "The contrasting fates..."

line 106: "emerging technologies to facilitate the administration of light deep within patients": please provide some background information on state of the art technologies (light guided devices, penetration depth of light within tissues as a function of wavelength, accessibility of tissues such as skin or inner organs)

Line 121: Section title is "Results and Discussion". Line 314: This section is entitled "Discussion". Should this rather be "Conclusion"?

Reviewer #2:

Remarks to the Author:

The manuscript reports on liposomes that modify their surface charge from neutral to cationic when they are exposed to light (365 nm) as an external, controllable stimulus. The authors have demonstrated both in vitro and in vivo (zebra fish) the feasibility and relevance of such a change in surface charge regarding liposomes circulation in blood and uptake by either endothelial cells or blood resident macrophages by adjusting the applied light dose. The introduction provides a clear picture of the state of the art and the relevance of the present study. The rationale behind the work is also clearly explained in the results and discussion section. The movies are also helpful for the understanding of the results. The experiments are described in detail.

Can the authors provide the pKa of the amino moieties generated after light irradiation? What is the expected percentage of positively charged moieties compared to the total of amino groups in each liposome? According to Figure 1, the amino group is always positively charged, which maybe not accurate at pH 7.4.

Lines 332-335. Our data indicates that above a certain cationic surface charge threshold, i.v. administered nanoparticles will predominantly “stick” to (and be internalised by) endothelial cells, particularly in blood vessels with reduced blood flow velocity, while below this charge threshold, nanoparticles will tend to aggregate in circulation and be subsequently cleared by the RES. Please define the threshold value and provide a hypothesis for aggregation.

The comment on the replacing 365 nm light with NIR light to irradiate profound organs lacks of analysis of feasibility. Please comment on the challenges of finding groups that can be readily broken under NIR light. Please check what value of wavelength is the correct (365 nm or 375 nm as stated in the supporting info).

Figure 6, plot g. The quality should be improved.

Do the authors record any increase in temperature during exposition of the liposomes to the 375 nm light?

Reviewer #3:

Remarks to the Author:

General

In the manuscript by Arias-Alpizar et al., the authors report on the development of liposomes of which the surface charge can be switched from neutral to cationic upon irradiation with UV light. The authors evaluate the photoactive liposomes in zebrafish models to determine the effects of surface charge switching on liposomal circulation time and uptake in endothelial cells and macrophages.

The paper is well written with appropriate literature references and results presented in a clear manner. The study's main feature is the demonstration of light-triggered switching of the liposome surface charge in vivo and imaging the effects in real time. While this proof-of-concept is of interest, the translational value and applicability of the approach can be subject to debate. This reviewer appreciates that the authors advocate for a thorough understanding of nano-bio interactions in vivo and argue that their approach shows that effective nanoparticles do not necessarily require complex designs. However, the ability to govern the biodistribution and cellular interactions of therapeutic nanoparticles (with straightforward composition) based on surface charge has been demonstrated by others, especially in the field of nucleic acid therapeutics (e.g. by Kranz et al. *Nature* 534, 396–401, 2016). In my opinion, the manuscript could be suitable for publication, provided several comments are addressed.

Specific

Specific

1. The authors mention in the Introduction section that based on their previous studies (reference 25) and experience, “anionic nanoparticles (<-20 mV measured zeta potential) interact strongly with RES cell types, namely scavenging endothelial cells (SECs, via a stabilin-mediated clearance pathway) and blood resident macrophages”. Have the authors also designed photocaged anionic lipids and determined liposomal cell uptake in zebrafish following irradiation?

2. The authors use fluorescently-labeled lipids or sulforhodamine B (SRB) as a fluorescent model payload to determine liposomal localization and cell uptake. These approaches provide qualitative rather than quantitative results and do not provide information if the delivered payload is functional. Would the authors be able to show functional payload delivery, for example using a reporter model such as Cre-Lox?

3. The authors mention in the final sentence of the Discussion section that “it is our hope that this study, and in particular the tools and methods employed, will expedite a transition from the empirical design of increasingly complex nanomedicines to the rational design of new, simple and effective nanomedicines”. How predictive/translational are the approach and the findings? It would strengthen the manuscript if this could be demonstrated. Would it for example be possible to irradiate (fluorescent) liposomal formulations prior to systemic administration in mice and determine cell uptake in macrophages and endothelial cells using flow cytometry compared to non-irradiated formulations?

Authors' responses to reviewers' comments for the manuscript:

Light-triggered switching of liposome surface charge directs the intracellular delivery of membrane impermeable payloads *in vivo*

Gabriela Arias-Alpizar, Li Kong, Redmar Vlieg, Alexander Rabe, Panagiota Papadopoulou, Michael S. Meijer, Sylvestre Bonnet, Stefan Vogel, John van Noort, Alexander Kros and Frederick Campbell

Firstly, could I thank all three reviewers for their careful consideration of our manuscript and their constructive comments. Given several comments relate to the translation of this technology into mammals, I will first comment, more generally, on the limitations of the technology, as described, as well as the translational value of the embryonic zebrafish. All specific comments are individually addressed below.

As described, the tissue penetration and cytotoxicity of UV-A light limit the translational potential of this proof-of-concept technology to larger (mammalian) models. Our choice of UV-A (370 nm) light was dictated by the use of *o*-nitrobenzyl (*o*-Nb) photocleavable functionality. These groups were chosen given their widespread use as photocage in complex chemical and biological systems (Ref. 32), their well characterised and clean (non-hydrolytic) photolysis, their synthetic accessibility and, importantly, their compatibility with fluorescent probes used to assess liposome biodistribution (by confocal microscopy) within small and transparent zebrafish embryos. Although strategies to improve access of UV light deep within tissue (e.g. injectable LED devices, Ref. 36); fiber optics, Refs. 34,35) are described, it is our view that translation of this technology to mice, as described, would be premature. We would instead advocate chemical modifications to the photocage to render liposomes sensitive to visible or near-infrared light. Photocages sensitive to single photon visible and near-IR light have recently been reported (Refs. 37-40). These views and additional references have been added to the conclusion (Lines 337-343).

In terms of translational value of the embryonic zebrafish, the adverse pharmacokinetics of cationic nanoparticles (e.g. DOPC:4→3 liposomes) are widely reported and our observations (*i.e.* non-specific adsorption to cells and serum aggregation) within the embryonic zebrafish are consistent with previous studies of cationic nanoparticles *in vitro* and *in vivo*. In terms of (near) neutral (e.g. DOPC:4 liposomes) and anionic nanoparticles, the zebrafish is an accurate, *qualitative* predictor of nanoparticle interactions with the mammalian RES (*i.e.* primarily LSECs and KCs within the liver; refs. 25, 26, 43). Up to 99% of systemically administered nanoparticles are cleared within the mammalian liver by these innate immune cells (Ref 6). Given the differences in relative cell number (SECs *vs* blood resident Mφs), RES tissue size and organization between the embryonic fish and adult mouse, however, it is not yet possible to make *quantitative* predictions of nanoparticle distribution across different RES cell types from embryonic fish to adult mouse. The embryonic zebrafish does not replace mouse experiments. Instead, it should be used to screen and assess fundamental *in vivo* clearance mechanisms of nanoparticles prior to first injections in higher vertebrates. These views and caveats have been added to the manuscript (Lines 312-318)

For photoactive, DOPC:4 (1:1) liposomes, the vast majority of liposomes remained freely circulating at one hour post-injection with no significant uptake evident within SECs and/or blood resident macrophages of the embryo. This mirrors the biodistribution of liposomes based on the lipid composition of Myocet® - a clinically approved liposomal-doxorubicin formulation with a known circulation lifetime of 2.5 h in humans. We are, therefore, confident that DOPC:4 (1:1) liposomes would demonstrate prolonged circulation lifetimes in mammals.

Responses to specific concerns and revisions made

COVID-19: Given the prolonged shutdown of our university, we would ask the reviewers to consider publication of this proof-of-concept work without additional experiments being performed. In instances where further experiments have been suggested, we have included additional text and references to try to strengthen claims made in the manuscript. See list of additional references at the end of this document.

Reviewer 1 comments

Comment: line 194: *Content leakage of liposomes was monitored over a time period of 30 minutes (Figure S6). What about stability over prolonged periods of time (e.g. storage at low temperature) of non-irradiated samples or irradiated samples during 24 hours?*

Author response: Further stability studies are undoubtedly necessary if this technology, as described, is to be taken beyond proof-of-concept in the embryonic zebrafish. However, as stated in the general comment above, we would first advocate further chemical modifications to render photoactive liposomes sensitive to longer wavelength, visible or NIR light (Refs. 37-39) to achieve tissue penetration depths of up to 2 cm (Ref. 40). Comprehensive stability/leakage studies should then be performed on these modified liposomes using pharmacologically active and clinically relevant payloads.

Comment: line 217: *"Having previously shown photoswitching of liposome surface charge occurred within seconds": please provide the reference*

Author response: This refers to *in vitro* photolysis experiments shown in Fig. 3b. The sentence (Line 206) now reads: "Having shown photoswitching of liposome surface charge occurs within seconds (Fig. 3b),..."

Comment: *Liposome aggregation in vivo due to changes in the protein corona before or after irradiation will trigger cellular uptake by macrophages. Were DLS experiments carried out in presence as well as absence of plasma proteins?*

Author response: Aggregation of cationic nanoparticles in serum is well described (Refs. 55,56). This is due to the adsorption of serum proteins (majority pI <7), and other (poly)anionic macromolecules, to the circulating nanoparticle surface. We have recently shown that cationic liposomes adsorb significantly more serum proteins than anionic or neutral liposomes (Ref. 57). Additional text and references have been added to the manuscript (Lines 231-234).

Comment: *Please explain the advantages and limitations of the used zebrafish model in more detail. Is it possible to extrapolate from data obtained in zebrafish to higher vertebrates?*

Author response: The translational potential of the embryonic zebrafish has been emphasized (see general comment above, Lines 316-322). Importantly, we have now stressed that the embryo only offers qualitative predictions of nanoparticle-RES interactions in higher vertebrates and does not replace experiments in more conventional animals.

Comment: Line 85: *"In our experience, anionic nanoparticles...": please provide the reference or does this relate to ref. 25?*

Author response: This refers to ref. 25. Citation has been added.

Comment: line 98: *"In the contrasting fates..." should read "The contrasting fates..."*

Author response: Text has been modified accordingly.

Comment: line 106: *"emerging technologies to facilitate the administration of light deep within patients": please provide some background information on state of the art technologies (light guided devices, penetration*

depth of light within tissues as a function of wavelength, accessibility of tissues such as skin or inner organs)

Author response: The following text has been added in the introduction (Lines 87-90): “...to apply light deep within patients. These include fiber-optic (Refs 34,35) and injectable microLED hardware (Ref 36), as well as photocleavable chemical functionality sensitive to visible or near infrared (NIR) light (Refs 37-39). Light wavelengths between 600 and 950 nm can penetrate various human tissues (skin, fat and blood) up to a depth of 2 cm (Ref 40).”

Comment: Line 121: Section title is "Results and Discussion". Line 314: This section is entitled "Discussion". Should this rather be "Conclusion"?

Author response: Section headed “Discussion” has been changed to “Conclusion”

Reviewer 2 comments

Comment: *Can the authors provide the pKa of the amino moieties generated after light irradiation? What is the expected percentage of positively charged moieties compared to the total of amino groups in each liposome? According to Figure 1, the amino group is always positively charged, which maybe not accurate at pH 7.4.*

Author response: Similar to aliphatic amines, the pKa of primary amines at PEG terminus is expected in the range 9–11 (see Xia, X. *et al.* Quantifying the Coverage Density of Poly(ethylene glycol) Chains on the Surface of Gold Nanostructures *ACS Nano*, **2012**, 6, 512-522). At physiological pH 7.4, it is reasonable to assume, therefore, that the majority (> 94%) of amines will be protonated. Given the small fraction of potentially uncharged amines present at the liposome surface (after photolysis), and the limited impact these would likely have on the overall surface charge, we feel it would be misleading to show uncharged amines in the Figure 1 schematic.

Comment: *Lines 332-335. Our data indicates that above a certain cationic surface charge threshold, i.v. administered nanoparticles will predominantly “stick” to (and be internalised by) endothelial cells, particularly in blood vessels with reduced blood flow velocity, while below this charge threshold, nanoparticles will tend to aggregate in circulation and be subsequently cleared by the RES. Please define the threshold value and provide a hypothesis for aggregation.*

Author response: In our experience, liposomes with a measured surface charge >20 mV tend to “stick” to endothelial cells, whereas those +5 to +20 mV tend to aggregate in circulation and be phagocytosed by blood resident macrophages of the embryonic zebrafish. While we believe a threshold cationic charge will apply to all classes of nanoparticles, this should be assessed on a case-by-case basis given the surface chemistry, self-assembled structure etc. of a nanoparticle will likely dictate the specific threshold value. These additional views have been added to the conclusion (Lines 333-337).

Aggregation of cationic liposomes is due to the adsorption of serum proteins (to the liposome surface (Refs. 55, 56). We have also recently shown that significantly more serum proteins absorb to the surface of cationic liposomes compared to anionic and neutral liposomes (Ref. 57). Additional text and references have been added to the manuscript to strengthen this point (Lines 231-234).

Comment: *The comment on the replacing 365 nm light with NIR light to irradiate profound organs lacks of analysis of feasibility. Please comment on the challenges of finding groups that can be readily broken under NIR light. Please check what value of wavelength is the correct (365 nm or 375 nm as stated in the supporting info).*

Author response: The main issue in finding groups that can be broken by (lower energy) visible or NIR light has been a lack of a structure–reactivity relationship to predict chemical structures primed for excited state heterolysis. However, guided by time-dependent density functional theory (TD-DFT) computational studies, a family of BODIPY-derived photocages has been recently reported that can be efficiently cleaved using visible or NIR light (Refs 37-39). In theory, zwitterionic BODIPY-photocages, connected to a cholesterylamine lipid anchor and incorporated within DOPC liposomes, should result in analogous surface charge switching, before and after light activation, as demonstrated for DOPC:4 and DOPC:4→3 liposomes, respectively.

The maximum measured wavelength of the LED was 370 ± 7 nm. This value is reported throughout the manuscript text. We have corrected the wavelength in Figure 1. The maximum wavelength reported by the manufacturer (Roithner Lasertechnik) for this LED is 375 nm. We have modified the SI text and figure caption S10 to try and make this discrepancy as clear as possible.

Comment: Figure 6, plot g. The quality should be improved.

Author response: This quality of this image has been improved. The resolution is 300 dpi.

Comment: Do the authors record any increase in temperature during exposition of the liposomes to the 375 nm light?

Author response: We have not measured any potential temperature increase of the embryo upon UV irradiation. Unfortunately, we can find no information relating to temperature increases due to UV-A irradiation of zebrafish embryos, however, based on human skin studies (albeit a significantly different system), a small temperature increase (1-4°C) of the embryo can be expected upon UV-A irradiation at the applied light doses (max. $\sim 80 \text{ J/cm}^2$) (see Kagetsu, N.; Gange, R.W.; Parrish, J.A. UV A-Induced Erythema, Pigmentation, and Skin Surface Temperature Changes Are Irradiance Dependent *J. Invest. Dermatol.* **1985**, 445-447).

Importantly however, while heat shock during early embryonic development in mammals generally results in deleterious consequences, the embryonic zebrafish (from 1dpf) is remarkably resilient to heat stress (Ref 51) and the maximum applied UVA light dose ($\sim 80 \text{ J/cm}^2$) is significantly below the reported LD₅₀ value (in embryonic zebrafish) of 850 J/cm^2 (Ref 50). We have added text (Lines 201-205) and references (Refs 50, 51) to the manuscript to highlight the suitability of the embryonic zebrafish for photoactivation studies using UV-A light.

Reviewer 3 comments

Comment: *The paper is well written with appropriate literature references and results presented in a clear manner. The study's main feature is the demonstration of light-triggered switching of the liposome surface charge in vivo and imaging the effects in real time. While this proof-of-concept is of interest, the translational value and applicability of the approach can be subject to debate. This reviewer appreciates that the authors advocate for a thorough understanding of nano-bio interactions in vivo and argue that their approach shows that effective nanoparticles do not necessarily require complex designs. However, the ability to govern the biodistribution and cellular interactions of therapeutic nanoparticles (with straightforward composition) based on surface charge has been demonstrated by others, especially in the field of nucleic acid therapeutics (e.g. by Kranz et al. Nature 534, 396–401, 2016).*

Author response: The novelty of our approach stems from our ability to drastically change the biodistribution of liposomes *in vivo* and on demand. Furthermore, the non-specific adsorption of cationic liposomes (upon light activation) means the *in vivo* target is not predetermined (in either time and/or space) and, in theory, any (vascularised) and light accessible tissue can be targeted.

Kranz et al. (*Nature* 534, 396–401, **2016**) describes varying lipoplex surface charge to achieve varying *in vivo* fates. The authors report lipoplexes with reduced cationic charge that largely accumulate in the spleen (and to a lesser extent the liver). As mentioned in the introduction, the spleen is also a RES organ and it is noteworthy that the highest uptake of lipoplexes was observed in splenic macrophages (although plasmid expression was highest in DCs). This result ties in with our observation that nanoparticles with reduced cationic surface charge are liable to aggregate in circulation (*via* serum protein adsorption) and are subsequently recognised and cleared by blood resident phagocytes.

Comment: *The authors mention in the Introduction section that based on their previous studies (reference 25) and experience, “anionic nanoparticles (<-20 mV measured zeta potential) interact strongly with RES cell types, namely scavenging endothelial cells (SECs, via a stabilin-mediated clearance pathway) and blood resident macrophages”. Have the authors also designed photocaged anionic lipids and determined liposomal cell uptake in zebrafish following irradiation?*

Author response: Given the extensive and rapid clearance of anionic nanoparticles by RES cell types in the embryonic zebrafish (Refs 25, 26), we have not considered applying our photo-switching methodology to anionic liposomes. We would be confident, however, that a liposome could be designed, analogous to the system described (e.g. using photocaged analogues of cholesteryl hemisuccinate), in which freely circulating liposomes could be directed to RES cell types on demand, using light as trigger.

Comment: *The authors use fluorescently-labeled lipids or sulforhodamine B (SRB) as a fluorescent model payload to determine liposomal localization and cell uptake. These approaches provide qualitative rather than quantitative results and do not provide information if the delivered payload is functional. Would the authors be able to show functional payload delivery, for example using a reporter model such as Cre-Lox?*

Author response: We fully acknowledge we do not show intracellular delivery of functional (and/or clinically relevant) payloads.

One option to demonstrate functional payload delivery would be to deliver a small molecule drug (e.g. doxorubicin) and monitor efficacy (e.g. tumor regression) in an appropriate disease model. In terms of payload, this approach would be most comparable to the demonstrated delivery of the small molecule dye, SR-B. Xenografting human cancer cells within the embryonic zebrafish is possible, and xenograft cancer cells can rapidly develop into tumors with recognisable pathophysiology within the embryonic fish (see Mione, M.C.; Trede, N.S. The zebrafish as a model for cancer *Dis. Model Mech.* **2010**, 517-523). However, given the small size of the embryo, the working timeframe (approx. 4 days) and the fact most small molecule drugs freely diffuse across lipid membranes, the relevance of assessing new drug delivery technologies in embryonic zebrafish disease models remains, in our view, highly contentious. Taken together with the fact we are unable to localise surface charge switching within the small embryo, we feel assessment of functional, small molecule drug delivery would only be appropriate in larger animals and more established disease models.

A more attractive approach, as the reviewer suggests, would be to monitor liposome-mediated delivery of functional biotherapeutics (e.g. proteins and RNA). Not only do these therapies exemplify the use of vector-based drug delivery systems, but their high endosomal degradation requires extensive assessment and optimisation to ensure functional payload delivery. Transgenic Cre reporting zebrafish lines are available (see Yoshikawa, S.; Kawakami, K.; Zhao, X.C. G2R Cre Reporter Transgenic Zebrafish *Dev. Dyn.* **2008**, 2460–2465), and although delivery of the Cre recombinase protein is conceivable, a more common approach is to deliver Cre mRNA to evaluate functional mRNA delivery and expression in target cells (e.g. Dahlman, J. E. *et al.* Barcoded nanoparticles for high throughput in vivo discovery of targeted therapeutics. *Proc. Natl. Acad. Sci.* **2017**, 2060–2065). For the *in vivo* delivery of exogenous mRNA (e.g. Cre mRNA), lipid nanoparticles (LNPs) have emerged as state-of-the-art non-viral vector (see Cullis, P. R.; Hope, M. J. Lipid Nanoparticle Systems for Enabling Gene Therapies. *Mol. Ther.* **2017**, 1467–1475). These technologies have been exemplified by the recent clinical approval of ONPATRO[®] (see Adams, D. *et al.* Patisiran, an RNAi therapeutic, for hereditary transthyretin amyloidosis. *N. Engl. J. Med.* **2018**, 11–21). LNPs are a distinct delivery vector to liposomes, however, and extending this work to photoswitchable LNPs, we feel, goes significantly beyond the scope of this proof-of-concept study.

Approaches to quantify intracellular payload delivery, such as single molecule tracking (e.g. Gilleron, J. *et al.* Image-based analysis of lipid nanoparticle-mediated siRNA delivery, intracellular trafficking and endosomal escape. *Nat. Biotechnol.* **2013**, 638–646) or signal amplification following cell isolation (e.g. Dahlman, J. E. *et al.* Barcoded nanoparticles for high throughput in vivo discovery of targeted therapeutics. *Proc. Natl. Acad. Sci.* **2017**, 2060–2065), are relevant for specific functional (and clinically relevant) payloads (particularly those that are susceptible to high endosomal degradation, e.g. RNA). These studies should be performed at a later stage of technological development using relevant payloads and are beyond the scope of this proof-of-concept study.

Comment: *The authors mention in the final sentence of the Discussion section that “it is our hope that this study, and in particular the tools and methods employed, will expedite a transition from the empirical design of increasingly complex nanomedicines to the rational design of new, simple and effective nanomedicines”. How predictive/translational are the approach and the findings? It would strengthen the manuscript if this could be demonstrated. Would it for example be possible to irradiate (fluorescent) liposomal formulations prior to systemic administration in mice and determine cell uptake in macrophages and endothelial cells using flow cytometry compared to non-irradiated formulations?*

Author response: See general response above. We believe strongly that the embryonic zebrafish can, and should, be used as an *in vivo* pre-clinical screen for qualitatively assessing and predicting nanoparticle interactions with key RES cell types, prior to first injections in rodents. In line with the principles of the 3Rs (Replacement, Reduction and Refinement) for the ethical use of animals in research, it is very much our hope that this study will encourage others to adopt the embryonic zebrafish within their nanomedicine development pipelines. As such, we are reluctant to extend this proof-of-concept study beyond this experimental organism. We are encouraged that these views are shared by the editor.

In terms of pre-irradiating liposomes prior to *i.v.* injection in rodents, we do not see the benefit of this over, for example, varying the mol% of cationic lipids and, therefore, cationic surface charge. As highlighted by the reviewer, the ability to redirect nanoparticles *in vivo* by fine tuning cationic surface charge has been elegantly described by e.g. Kranz *et al.* (*Nature* 534, 396–401, **2016**) and is also demonstrated in this work (Figure 2d–m)

List of additional references

Ref 26 – Hayashi, Y. *et al.* Differential Nanoparticle Sequestration by Macrophages and Scavenger Endothelial Cells Visualized in Vivo in Real-Time and at Ultrastructural Resolution. *ACS Nano* **14**, 1665–1681 (2020)

- **Reason for inclusion:** *A very recent paper describing the dynamic uptake of anionic nanoparticles by SECs and blood resident macrophages in the embryonic zebrafish.*

Ref 34 – Moore, C. M., Pendse, D. & Emberton, M. Photodynamic therapy for prostate cancer - A review of current status and future promise. *Nat. Clin. Pract. Urol.* **6**, 18–30 (2009).

- **Reason for inclusion:** *A review of clinical PDT for prostate cancer with particular focus on the use of fibre optics to deliver light deep within patients*

Refs 37-39 – (37) Rubinstein, N., Liu, P., Miller, E. W. & Weinstain, R. Meso-Methylhydroxy BODIPY: A scaffold for photo-labile protecting groups. *Chem. Commun.* **51**, 6369–6372 (2015). (38) Goswami, P. P. *et al.* BODIPY-Derived Photoremovable Protecting Groups Unmasked with Green Light. *J. Am. Chem. Soc.* **137**, 3783–3786 (2015). (39) Peterson, J. A. *et al.* Family of BODIPY Photocages Cleaved by Single Photons of Visible/Near-Infrared Light. *J. Am. Chem. Soc.* **140**, 7343–7346 (2018).

- **Reason for inclusion:** *These papers describe the discovery and characterisation of BODIPY-derived photocages that are sensitive to single photon visible or NIR light.*

Ref 40 – Smith, A. M., Mancini, M. C. & Nie, S. Bioimaging: Second window for in vivo imaging. *Nat. Nanotechnol.* **4**, 710–711 (2009).

- **Reason for inclusion:** *This News and Views article clearly describes the optical windows and tissue penetration depths of light in various human tissues.*

Ref 43 – Sieber, S. *et al.* Zebrafish as a predictive screening model to assess macrophage clearance of liposomes in vivo. *Nanomedicine Nanotechnology, Biol. Med.* **17**, 82–93 (2019).

- **Reason for inclusion:** *This paper describes the use of the embryonic zebrafish as a screening tool to predict nanoparticle uptake in (blood resident) macrophages of higher vertebrates.*

Refs 50-51 – (50) Dong, Q., Svoboda, K., Tiersch, T. R. & Todd Monroe, W. Photobiological effects of UVA and UVB light in zebrafish embryos: Evidence for a competent photorepair system. *J. Photochem. Photobiol. B Biol.* **88**, 137–146 (2007). (51) Wang, R., Zhang, H., Du, J. & Xu, J. Heat resilience in embryonic zebrafish revealed using an in vivo stress granule reporter. *J. Cell Sci.* **132**, jcs234807 (2019)

- **Reason for inclusion:** *These papers exemplify the suitability of the embryonic zebrafish for photoactivation studies.*

Refs 53-55 – (53) Rausch, K., Reuter, A., Fischer, K. & Schmidt, M. Evaluation of nanoparticle aggregation in human blood serum. *Biomacromolecules* **11**, 2836–2839 (2010). (54) Zhao, W., Zhuang, S. & Qi, X. R. Comparative study of the in vitro and in vivo characteristics of cationic and neutral liposomes. *Int. J. Nanomedicine* **6**, 3087–3098 (2011). (55) Pattipeiluhu, R. *et al.* Unbiased Identification of the Liposome Protein Corona using Photoaffinity-based Chemoproteomics. *ACS Cent. Sci.*, acscentsci.9b01222 (2020).

- **Reason for inclusion:** *These papers are included to highlight extensive serum protein adsorption to the surface of cationic nanoparticles. This causes aggregation of nanoparticles in circulation and subsequent recognition and clearance by blood resident macrophages.*

Reviewers' Comments:

Reviewer #1:

Remarks to the Author:

I thank the authors for addressing my questions and concerns adequately. I recommend publication. I have no further requests.

Reviewer #2:

Remarks to the Author:

The manuscript has been improved.

Reviewer #3:

Remarks to the Author:

In my opinion, the authors have addressed all reviewer comments to a satisfactory extent. I consider the manuscript suitable for publication in Nature Communications.

During the peer review process, the following study was published by Dan Siegwart et al. in Nature Nanotechnology: Cheng, Q., Wei, T., Farbiak, L. et al. Selective organ targeting (SORT) nanoparticles for tissue-specific mRNA delivery and CRISPR–Cas gene editing. *Nat. Nanotechnol.* 15, 313–320 (2020). <https://doi.org/10.1038/s41565-020-0669-6>.

This study shows tissue-specific (liver, spleen, lungs) delivery of functional mRNA based on lipid nanoparticle surface charge. Although it concerns different nanoparticles, perhaps it is of interest to mention in the Introduction or Discussion section.